# Recursive Models for Long-Horizon Reasoning

Chenxiao Yang [1]   Nathan Srebro [1]   Zhiyuan Li [1]

## Abstract

Modern language models reason within bounded context, an inherent constraint that poses a fundamental barrier to long-horizon reasoning. We identify recursion as a core principle for overcoming this barrier, and propose recursive models as a minimal realization, where the model can recursively invoke itself to solve subtasks in isolated contexts. We prove that any computable problem admits a recursive decomposition of reasoning in which each subtask requires only exponentially smaller active context than standard autoregressive models; this strictly surpasses any context management approach confined to a single sequence, such as summarization. We further generalize our framework to modern agentic systems with arbitrary context processing and control flows, and prove that recursive models can achieve optimal power within this broader class. Experimentally, we train a 3B model to reason recursively and evaluate on Boolean satisfiability, a task requiring long-horizon combinatorial search, where it significantly outperforms frontier LLMs.

## 1. Introduction

Modern language models exhibit remarkable general problem solving power (Radford et al., 2018; 2019; Brown et al., 2020; Achiam et al., 2023). Through extended thinking (Wei et al., 2022; OpenAI, 2024; Guo et al., 2025) and agentic systems (Yao et al., 2023; Shinn et al., 2023; Park et al., 2023), they can handle increasingly complex tasks across diverse domains. Nevertheless, these systems are subject to a physical constraint: at every step, the model can only attend to bounded-sized context window, strictly limiting what can be computed in a single forward pass.

This has driven growing interest in effective context management. For instance, summarization compresses lengthy

reasoning traces into compact states, discarding no longer needed history to free up space (Yang et al., 2025b; Yu et al., 2025; Zhou et al., 2025; Yan et al., 2025); memory-augmented approaches write and retrieve relevant information in external storage (Packer et al., 2024; Chhikara et al., 2025; Suzgun et al., 2025; Xu et al., 2025); and in agentic systems, subtasks are distributed across agents, each operating in its own context while collaborating toward a shared goal (Hong et al., 2024; Wu et al., 2023; Li et al., 2023).

Yet questions remain: how do these different systems formally compare in their reasoning power? What core mechanisms, as scaffolding that wraps around the base generator, can enable models to handle long-horizon tasks that are otherwise impossible because of context constraints? And are these mechanisms optimal? Despite the importance of these questions, existing work lacks a formalization for these questions to be answered systematically. Notable related works are Yang et al. (2025b;c), which, however, focus on summarization-based context management and self-correction in diffusion language models respectively.

In this work, we identify recursion as a core principle for overcoming context constraints, and a form of computational power naturally enabled by modern agentic systems. In a broad sense, recursion refers to the application of a finite, static set of rules to a target problem, that dynamically produces a potentially infinite depth of behaviors that, though contextually isolated from each other, contribute to the final solution.

We propose the simplest realization of this principle, which we call recursive model. It consists of a single base LLM as the generator, equipped with two minimal tools, `call` and `return`. As illustrated in Figure 1c, the model can invoke itself: `call` creates an isolated context and the model solves the subtask there independently; upon completion, `return` discards the intermediate reasoning and passes only the final answer back to the parent context. Since each invoked model can itself invoke further calls, this enables a deep context stack while keeping each individual context bounded by the maximal context length. Similar ideas have been explored in earlier and concurrent work (Lee & Kim, 2023; Prasad et al., 2024; Schroeder et al., 2025; Pan et al., 2025; Zhang et al., 2025c; Sun et al., 2025; Zhang et al., 2025a); see a comprehensive discussion in § A.

[1]Toyota Technological Institute at Chicago. Correspondence to: Chenxiao Yang <chenxiao@ttic.edu>.

*Proceedings of the 43rd International Conference on Machine Learning*, Seoul, South Korea. PMLR 306, 2026. Copyright 2026 by the author(s).

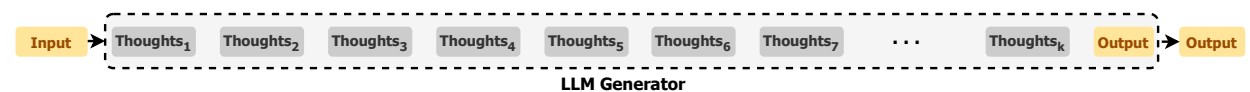

*(a)* **Standard Autoregressive Model.** The model generates tokens sequentially, appending each to the current sequence until the context limit is reached.

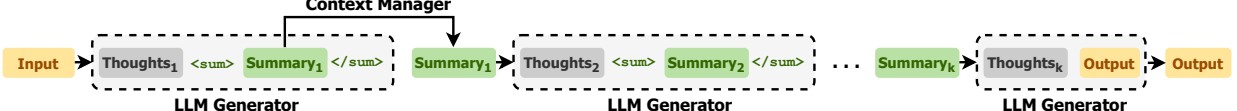

*(b)* **Single-Context Model.** The entire generation process operates within a single sequence. As a representative example, summarization periodically compresses past reasoning into a compact summary and discards the original tokens.

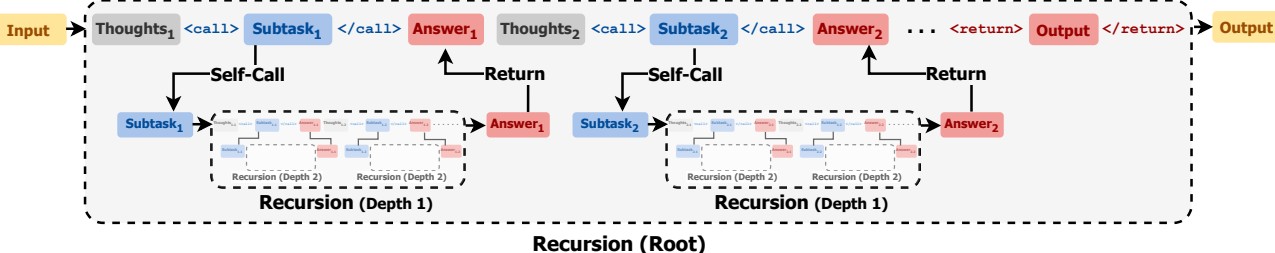

*(c)* **Recursive Model.** Unlike the previous two approaches, computation spans multiple isolated contexts. The model delegates subtasks via `call`, each solved in a fresh context; `return` passes back only the result, discarding intermediate reasoning. This enables unbounded recursion depth without growing any single context.

*Figure 1.* Overview of different context management strategies.

One important observation is that the recursive model naturally induces a separation between local and global space: the generator only needs to attend to the active context, while inactive contexts in the context stack can be offloaded to external storage and restored upon return. While this improves space efficiency, it seems to impose a strong requirement that problems must admit modular decompositions. Do general computational problems possess such structure? We show the answer is affirmative: any computable problem inherently admits a recursive decomposition, and furthermore, by doing so, the required context can be reduced exponentially. Specifically, we prove that with local space $S(n)$, recursive models can solve any problem requiring up to $\exp(\mathcal{O}(S(n)))$ computation time. In comparison, standard autoregressive models would require context length $\exp(\mathcal{O}(S(n)))$ to solve the same problems, which is an exponential gap.

Recursion, however, is not the only approach for context management. Consider summarization (Figure 1b), which periodically compresses the context and discards old history to keep the context window bounded. Unlike recursion, summarization and indeed most existing strategies keep the entire generation process within a single sequence. We call these single-context models. Prior work (Yang et al., 2025b) shows that with context length $S(n)$, summarization can solve all problems requiring $S(n)$ space. We prove that this is in fact optimal: no single-context model, regardless of its context management strategy, can surpass summarization, which is, however, still strictly less powerful than recursion.

Indeed, we show that even constant-depth recursion (i.e., depth 1) suffices to match the optimum of all single-context models. Moreover, deeper recursion breaks through this ceiling, solving problems beyond what any single-context approach can reach. This separates the power of recursive models from those shallow counterparts (Sun et al., 2025; Zhang et al., 2025a).

Modern agentic systems are unique in that they are no longer confined to a single context: they can dynamically spawn contextually isolated sub-agents to solve specialized subtasks independently, and the responses are integrated back, processed, and used to determine the system's next behavior. This unique feature enables recursion in broader use cases. While not all agentic systems possess this capability, we formalize a powerful family called recursive agentic systems, which equip agentic systems with scaffoldings that create a recursive control loop. The recursive model is the minimal realization of this family. We show that any agentic system that is recursive can reach the same power as recursive models, enabling them to break through context constraints far beyond standard approaches. Yet, none can surpass recursive models, suggesting that the recursive model, despite its simplicity, is already optimally powerful within this family.

Experimentally, we train Qwen2.5-3B-Instruct to perform recursive reasoning on SAT, a canonical NP-complete problem that naturally admits recursive decomposition via backtracking search. The resulting model significantly outperforms non-recursive baselines while requiring significantly smaller context.

---

**Algorithm 1** Autoregressive Generator, $f^{\mathrm{cot}}$

---

**Input:** Input sequence $\mathbf{x} \in \Sigma^*$, next-token generator $\pi :$
 $\Sigma^* \to \Sigma$, stopping condition stop $: \Sigma^* \to \{0, 1\}$.
 1: **while** $\neg\mathrm{stop}(\mathbf{x})$ **do**
 2:  Generate $y \leftarrow \pi(\mathbf{x})$
 3:  Append $\mathbf{x} \leftarrow \mathbf{x} \parallel y$
 4: **return x**

---

**Algorithm 2** Recursive Model, $f^{\mathrm{rm}}$

---

**Input:** $\mathbf{x} \in \Sigma^*$; sequence generator $f : \Sigma^* \rightharpoonup \Sigma^*$ whose
 defined outputs end with a call or return string.
 1: **while true:**
 2: $\mathbf{y} \leftarrow f(\mathbf{x})$
 3: **if** $\mathbf{y} = \mathbf{y}' \parallel \langle\texttt{return}\rangle\mathbf{a}\langle/\texttt{return}\rangle$: **return a**
 4: **if** $\mathbf{y} = \mathbf{y}' \parallel \langle\texttt{call}\rangle\mathbf{q}\langle/\texttt{call}\rangle$: $\mathbf{x} \leftarrow \mathbf{y}' \parallel f^{\mathrm{rm}}(\mathbf{q})$

---

## 2. Recursive Models

This section defines the recursive model. The construction takes a partial sequence generator $f : \Sigma^* \rightharpoonup \Sigma^*$, which maps a prompt, or more generally a context, to a generated sequence. Our default choice is the CoT sequence generator $f^{\mathrm{cot}}$, obtained by autoregressive rollout from a next-token generator.

**Autoregressive Generator.** Let $\pi : \Sigma^* \to \Sigma$ be a next-token generator and let stop $: \Sigma^* \to \{0, 1\}$ be a stopping condition. Algorithm 1 defines the partial sequence generator $f^{\mathrm{cot}} : \Sigma^* \rightharpoonup \Sigma^*$. Starting from an input sequence $\mathbf{x}$, the rollout repeatedly appends the token $\pi(\mathbf{x})$ until $\mathrm{stop}(\mathbf{x}) = 1$, then returns the final sequence. If stop never holds, $f^{\mathrm{cot}}(\mathbf{x})$ is undefined. We write $\mathbf{x} \parallel \mathbf{z}$ for sequence concatenation.

**Recursive Model.** Fix a partial sequence generator $f : \Sigma^* \rightharpoonup \Sigma^*$; by default, $f = f^{\mathrm{cot}}$. We define the recursive model induced by $f$ as a function $f^{\mathrm{rm}} : \Sigma^* \rightharpoonup \Sigma^*$. We suppress the dependence on $f$ in the notation. Its input $\mathbf{x}$ is the full root prompt, and its output, when defined, is the answer returned by the root context. Execution starts from the root stack $\mathbf{S}_0 = [\mathbf{x}]$.

For $t = 0, 1, \ldots$, $\mathbf{S}_t$ denotes the stack after $t$ stack updates; each update happens after one complete call to $f$, not after one generated token in Algorithm 1. Each stack $\mathbf{S}_t \in (\Sigma^*)^+$ is a non-empty list of token sequences. Only the top sequence is active: it is the full input passed to $f$ at stack update $t$. The lower sequences $\mathbf{S}_t[: -1]$ are suspended parent contexts that are not visible to $f$ until control returns to them. For a stack $\mathbf{S}$ and sequences $\mathbf{s}_1, \ldots, \mathbf{s}_k$, $\mathrm{Push}(\mathbf{S}; \mathbf{s}_1, \ldots, \mathbf{s}_k)$ appends them to $\mathbf{S}$ in order.

The recursive model uses four reserved delimiter tokens $\langle\texttt{call}\rangle, \langle/\texttt{call}\rangle, \langle\texttt{return}\rangle, \langle/\texttt{return}\rangle \in \Sigma$. We write $\langle\texttt{call}\rangle\mathbf{q}\langle/\texttt{call}\rangle$ and $\langle\texttt{return}\rangle\mathbf{a}\langle/\texttt{return}\rangle$ for the delimited call and return strings. When the default choice $f = f^{\mathrm{cot}}$ is used, the stopping condition in Algorithm 1 is chosen so that $f^{\mathrm{cot}}$ returns only when the current sequence ends with one of these strings.

A call pauses the current context and starts a new child context for the subproblem. A non-root return removes the child context and appends only its answer to the parent; the child's intermediate tokens are not copied back. Formally, at stack update $t$, run $f$ on the active context and let $\mathbf{y}_t := f(\mathbf{S}_t[-1])$. The generator first produces the full sequence $\mathbf{y}_t$; only then do we parse its final call or return. We represent this intermediate state by replacing the old stack top with $\mathbf{y}_t$, and then apply the stack update rule:

$$\widetilde{\mathbf{S}}_t := \mathrm{Push}(\mathbf{S}_t[: -1]; \mathbf{y}_t), \qquad \mathbf{S}_{t+1} = \mathrm{Step}(\widetilde{\mathbf{S}}_t). \quad (1)$$

Thus $\widetilde{\mathbf{S}}_t$ is the stack after the generator output is produced, while $\mathbf{S}_{t+1}$ is the stored stack after the call or return is processed. The map Step is defined on a transient stack $\widetilde{\mathbf{S}}$ by two cases. Write $\mathbf{y} := \widetilde{\mathbf{S}}[-1]$ for its top sequence.

$$\begin{aligned}
\textsc{Call:}\ &\mathrm{Step}(\widetilde{\mathbf{S}}) = \mathrm{Push}(\widetilde{\mathbf{S}}[: -1]; \mathbf{y}', \mathbf{q}) \\
&\quad \text{if } \mathbf{y} = \mathbf{y}' \parallel \langle\texttt{call}\rangle\mathbf{q}\langle/\texttt{call}\rangle; \\
\textsc{Return:}\ &\mathrm{Step}(\widetilde{\mathbf{S}}) = \mathrm{Push}(\widetilde{\mathbf{S}}[: -2]; \widetilde{\mathbf{S}}[-2] \parallel \mathbf{a}) \\
&\quad \text{if } \mathbf{y} = \mathbf{y}' \parallel \langle\texttt{return}\rangle\mathbf{a}\langle/\texttt{return}\rangle \\
&\quad \text{and } |\widetilde{\mathbf{S}}| > 1.
\end{aligned}$$
$$(2)$$

A root return is not a stack update. If $|\widetilde{\mathbf{S}}_t| = 1$ and $\widetilde{\mathbf{S}}_t[-1] = \mathbf{y}' \parallel \langle\texttt{return}\rangle\mathbf{a}\langle/\texttt{return}\rangle$, the computation terminates and returns $\mathbf{a}$. The recursive model is partial: if some call to $f$ is undefined, if $f$ returns a sequence that matches neither form above, if a child computation is undefined, or if execution never reaches a root return, then $f^{\mathrm{rm}}(\mathbf{x})$ is undefined.

Algorithm 2 gives an equivalent recursive view of the same process, showing only the active context $\mathbf{x}$. In the call case, $\mathbf{x} \leftarrow \mathbf{y}' \parallel f^{\mathrm{rm}}(\mathbf{q})$ means: pause the parent after $\mathbf{y}'$, solve the child prompt $\mathbf{q}$ using the same sequence generator $f$, append the child's returned answer, and continue in the parent.

In the experiments, we run this model with a finite iteration budget; § B.3 gives the exact procedure.

### 2.1. Variants and Extensions

The basic recursive model above is our default. We will also use two variants that change only what information is visible in a context, while leaving the meaning of call and return unchanged.

**Variant 1: Prompt Prefixing.** Some constructions need every child call to see the original problem instance. Let $\mathbf{x}_0$ denote the root prompt. Instead of copying $\mathbf{x}_0$ into every generated subproblem, we expose it as a fixed prefix whenever the active context is non-root:

$$\mathbf{x}_0 \parallel \mathbf{y}_t = f(\mathbf{x}_0 \parallel \mathbf{S}_t[-1]), \quad \text{when } |\mathbf{S}_t| > 1. \quad (3)$$

Here $\mathbf{y}_t$ is the generator output after removing the fixed prefix $\mathbf{x}_0$; for the default CoT rollout $f^{\text{cot}}$, this prefix is always present because rollout only appends tokens to its input. The stack update itself is still the ordinary transition in Equation (1). Root-level steps do not add this prefix.

**Variant 2: Question Preservation.** In the basic call rule, the parent keeps only $\mathbf{y}'$ while the child receives $\mathbf{q}$ as its complete prompt. Thus, after the child returns, the parent sees the answer but not the subtask text. To keep the subtask text in the parent as well, replace the call rule by

$$\mathbf{S}_{t+1} = \mathsf{Push}(\mathbf{S}_t[:-1]; \mathbf{y}' \parallel \mathbf{q}, \mathbf{q}). \quad (4)$$

This changes only what the parent remembers; the child prompt is still $\mathbf{q}$.

**Further Extensions.** § 4 formalizes a more general model in which the fixed stack-transition rule is replaced by a scaffold. Such a scaffold may parse model outputs, add instructions, call tools or other generators, and decide when to launch a recursive call. We call the resulting systems recursive agentic systems. Unless a variant or extension is explicitly invoked, all results use the basic recursive model above.

# 3. Computational Power of Recursive Models

Recursive calls organize complex tasks as nested subcomputations. The key resource question is simple: the full stack may be large, but the generator sees only the top context at any moment. We therefore measure both the total stack size and the largest active context, and ask how much power is gained by allowing deep recursion rather than forcing all reasoning into one sequence.

## 3.1. Separation of Global and Local Spaces

Unlike the standard generation process where context grows monotonically, recursive models work on a stack of sequences, which gives rise to two natural resource measures:

**Definition 1** (Global and Local Space)**.** For a stack $\mathbf{S}$, we define the *global space* and *local space* respectively as:

$$\mathsf{GS}(\mathbf{S}) := \sum_{\mathbf{s} \in \mathbf{S}} |\mathbf{s}|, \qquad \mathsf{LS}(\mathbf{S}) := \max_{\mathbf{s} \in \mathbf{S}} |\mathbf{s}|, \quad (5)$$

where global space $\mathsf{GS}$ refers to the total number of tokens across all sequences, and local space $\mathsf{LS}$ refers to the length of the longest sequence.

This resource distinction is practically significant: the **global space** corresponds to the total size of the current stack (including suspended and active contexts). Suspended contexts (i.e., all but the stack top) are temporarily inactive and can be stored outside the active attention window as text, token sequences, or KV caches, so storage size and transfer latency are implementation costs rather than part of the local-space measure.

In contrast, **local space** is the maximum length of the active context window throughout next-token generation. Unlike suspended contexts, the active context must fit within the model's attention window during each generator call, making local space the practical bottleneck. **We thus focus our analysis on the reasoning power achievable under strict local space constraints.**

**Base Transformer Model.** For the complexity results, the next-token generator is a constant-size causal Transformer with average-hard attention, as formalized in § C. Autoregressive rollout turns this generator into the CoT sequence function from § 2, and the stack update rule then turns that sequence function into a recursive model.

**RM Complexity Class.** We use RM for language classes, and $f^{\text{rm}}$ for an individual recursive model. A fixed recursive model decides a language if, on every input, the root context returns a designated accept or reject symbol. The class $\mathsf{RM}(S(n), D(n), T(n))$ records what can be decided when the active context length, recursion depth, and total number of generated tokens are bounded by $S(n)$, $D(n)$, and $T(n)$.

**Definition 2** (Recursive Model Complexity Class)**.** For functions $S, D, T : \mathbb{N} \rightarrow \mathbb{N}$, the class $\mathsf{RM}(S(n), D(n), T(n))$ consists of all decision problems solvable by recursive models obtained from constant-size, $\mathcal{O}(\log S(n))$-precision Transformers as above, such that for all inputs $\mathbf{x} \in \Sigma^n$:

1. **Local Space**: $\max_t \mathsf{LS}(\widetilde{\mathbf{S}}_t) \leq S(n)$, where $t$ ranges over completed generator calls, including the final root-return call;

2. **Recursion Depth**: $\max_t |\mathbf{S}_t| \leq D(n)$ (the stack depth is bounded by $D(n)$);

3. **Total Steps**: the total number of generated tokens is at most $T(n)$.

Here $S(n)$ bounds the total length of one active context, including the original input or prompt prefix whenever it is visible to that call. When no time constraint is imposed, we write it as $\mathsf{RM}(S(n), D(n))$.

**Standard Complexity Classes.** To characterize the expressivity of recursive models, we compare with standard Turing machine complexity classes. We denote by $\mathsf{TIME}(T(n))$ and $\mathsf{SPACE}(S(n))$ the classes of problems

solvable in $T(n)$ time and $S(n)$ space, respectively. We write $\mathsf{TM}(S(n), T(n))$ for the simultaneous class of languages decided by a deterministic Turing machine that uses $\mathcal{O}(S(n))$ space and $\mathcal{O}(T(n))$ time on all inputs. (See § D for formal definitions.)

### 3.2. Main Result

Now we formally establish the computational power of recursive models with unbounded recursion depth.

**Theorem 1** (Deep Recursive Models). *For any $S(n) \geq n$, recursive models can solve any problem in $\mathsf{TIME}(2^{\mathcal{O}(S(n))})$ under local space constraint $\mathcal{O}(S(n))$:*

$$\mathsf{TIME}(2^{\mathcal{O}(S(n))}) \subseteq \mathsf{RM}(\mathcal{O}(S(n)), \infty, \infty). \quad (6)$$

The theorem is about active working context: exponentially long computations can be organized into many small frames. At each step, the generator attends only to the current frame, while suspended frames are stored outside the active attention window.

The proof gives a more explicit form: each active context stores the input plus $\mathcal{O}(\log T(n))$ auxiliary tokens for indexing the simulated time step and tape position. Thus, for $T(n) = 2^{\mathcal{O}(s(n))}$, the simulation uses local context $n + \mathcal{O}(s(n))$:

$$\mathsf{TIME}(2^{\mathcal{O}(s(n))}) \subseteq \mathsf{RM}(n + \mathcal{O}(s(n)), \infty, \infty). \quad (7)$$

**Remark 1: Input versus working memory.** This distinction matters when working memory is much smaller than the input length. Recursion does not make the active context shorter than the input tokens a call must read. For tasks such as Needle-in-a-Haystack, where the answer is hidden in a long input, the bottleneck is access to the long input; recursion helps only after the needed information is in the active context, or if the model has another way to retrieve the relevant input tokens.

**Remark 2: Depth and runtime.** Achieving this simulation may require recursion depth $2^{\mathcal{O}(s(n))}$. Without memoization, repeated subcalls may inflate the total number of generated tokens; memoization can reduce this overhead, but is not part of the basic recursive model. We provide two proofs in § E and § F: the first expresses Turing machine computation as recursive functions, and the second uses the classical alternating-space characterization of exponential time (Arora & Barak, 2009). The above results also apply to the two variants discussed in § 2.1.

### 3.3. No Recursion and Shallow Recursion

Next, we show that the depth of recursion is critical to the power of recursive models: without deep recur-

sion, the model is no more powerful than simpler context-management approaches.

**Standard Autoregressive Models.** When $D(n) = 1$, no recursive calls are made and the model reduces to standard autoregressive models (a.k.a. CoT). While it is known that with sufficiently many intermediate steps, autoregressive models can solve any computable problem (Merrill & Sabharwal, 2024; Feng et al., 2024; Li et al., 2024; Yang et al., 2025b), this comes at a significant cost:

**Theorem 2** (Standard Autoregressive Models / CoT). *For a standard autoregressive model (i.e., recursive model with depth $D = 1$) with local space $\mathcal{O}(S(n))$, $S(n) \geq n$, we have:*

$$\mathsf{TIME}(\mathcal{O}(S(n))) \subseteq \mathsf{RM}(\mathcal{O}(S(n)), 1), \quad (8)$$

$$\mathsf{RM}(\mathcal{O}(S(n)), 1) \subseteq \mathsf{TIME}(\widetilde{\mathcal{O}}(S^2(n))). \quad (9)$$

Both inclusions follow from Merrill & Sabharwal (2024) (Eq. (1)); the $\widetilde{\mathcal{O}}(\cdot)$ absorbs the polylogarithmic overhead of simulating $\mathcal{O}(\log S(n))$-precision arithmetic on a Turing machine. Together, the two inclusions show that standard autoregression with context length $S(n)$ (which determines the total reasoning steps when $D(n) = 1$) can solve all problems in $\mathsf{TIME}(\mathcal{O}(S(n)))$, but its power is contained in $\mathsf{TIME}(\widetilde{\mathcal{O}}(S^2(n)))$.

Compared with Theorem 1, this gives an exponential saving in local context for these long computations: solving the same exponential-time class without recursion would require exponentially larger context. For instance, with polynomial context $S(n) = \mathrm{poly}(n)$, standard models are confined to P, while recursive models reach EXPTIME, which is beyond NP and PSPACE under standard assumptions.

**Constant-Depth Recursion.** Constant recursion depth already improves over plain autoregression, but only up to the power of single-context management strategies such as summarization:

**Theorem 3** (Constant-Depth Recursive Models). *For any $S(n) \geq n$, recursive models with constant recursion depth $D = O(1)$ and local space $\mathcal{O}(S(n))$ can solve any problem in $\mathsf{SPACE}(S(n))$:*

$$\mathsf{SPACE}(S(n)) \subseteq \mathsf{RM}(\mathcal{O}(S(n)), \mathcal{O}(1)). \quad (10)$$

*More generally, the same construction preserves the time bound of such a simultaneous space-time simulation:*

$$\mathsf{TM}(S(n), T(n)) \subseteq \mathsf{RM}(\mathcal{O}(S(n)), \mathcal{O}(1), \mathcal{O}(T(n))). \quad (11)$$

This result shows that constant-depth recursion achieves both space and time efficiency relative to a space-$S(n)$, time-$T(n)$ computation: the local space matches the actual

space complexity $S(n)$, and the total number of generated tokens matches the time complexity $T(n)$. The proof uses tail-recursive simulations; see § E for details and for the caveat about the question-preservation variant.

However, this does not exceed optimal single-context management. This computational power matches that of **summarization** (Yang et al., 2025b), which periodically compresses reasoning history to free up space (illustrated in Figure 1b). In fact, as we will prove later (§ 4), SPACE($S(n)$) is the *maximum* expressive power single-context management can achieve, **and constant-depth recursion therefore offers no advantage over single-context management strategies**.

Yet even this upper bound is exponentially weaker than deep recursion: comparing with Theorem 1, there is a gap from SPACE($S(n)$) to TIME($2^{\mathcal{O}(S(n))}$). For polynomial context $S(n) = \text{poly}(n)$, this is the gap between PSPACE and EXPTIME, widely believed to be strict.

# 4. Generalization to Agentic Systems

While the recursive model in § 2 uses a minimal fixed controller that updates a stack according to the generator's calls and returns, real agentic systems can use richer fixed controllers around LLMs, tools, and specialized agents (Gao et al., 2025; Wang et al., 2024; Hong et al., 2024; Wu et al., 2023). This section formalizes this more general model and asks whether richer controllers are more powerful under the same local-space bound.

## 4.1. Formalizing Recursive Agentic Systems

We call such a controller a *scaffold*. Given an input string, a scaffold maintains the text of one run, such as the current prompt, scratch work, and parsed fields. It chooses which strings to send to generators or tools, uses the returned strings to update the run, and may solve a subproblem by starting another scaffold run on a new input string. The caller resumes when that run returns. Formally:

**Definition 3** (Recursive Agentic System). A *recursive agentic system* is a pair $(\mathcal{S}, \mathcal{F})$ consisting of:

1. generators $\mathcal{F} = (f_1, \ldots, f_k)$, where each $f_\ell : \Sigma^* \to \Sigma^*$ models a language model or string-valued tool;

2. scaffolds $\mathcal{S} = (S_1, \ldots, S_m)$, where each $S_i$ is a deterministic controller with string input and, when it halts, string output. During execution, $S_i$ may issue queries $\text{GEN}_\ell(u)$ with $\ell \in [k]$ or $\text{SELF}_j(u)$ with $j \in [m]$, where $u \in \Sigma^*$.

These queries are interpreted as follows. A query $\text{GEN}_\ell(u)$ sends $u$ to generator $f_\ell$ and returns $f_\ell(u)$. A query $\text{SELF}_j(u)$ starts a new run of scaffold $S_j$ on input $u$; if that run returns a string $z$, the caller receives $z$ and contin-

ues. Between queries, the scaffold's control is deterministic: it may update its stored text, halt with a string output, issue another query, or continue running. § I.3 formalizes this controller as an oracle Turing machine variant with output, which gives a standard way to measure local workspace and query strings in the resource bounds below.

**Induced functions.** The system induces one partial function for each scaffold:

$$\phi_i^{\mathcal{S},\mathcal{F}} : \Sigma^* \rightharpoonup \Sigma^*, \qquad i \in [m]. \tag{12}$$

The function $\phi_i^{\mathcal{S},\mathcal{F}}$ is the partial input-output function obtained by starting scaffold $S_i$ with the generators $\mathcal{F}$. Thus $\phi_i^{\mathcal{S},\mathcal{F}}(x) = y$ exactly when the complete run of $S_i$ on input $x$ terminates with output $y$, with GEN queries answered by the corresponding $f_\ell$ and SELF queries evaluated as recursive scaffold runs. Every recursive call made during this run must itself return; if the root run diverges, or if some required recursive call never returns, then $\phi_i^{\mathcal{S},\mathcal{F}}(x)$ is undefined. In the oracle Turing machine formalization of § I.3, this is the corresponding non-halting computation. Since recursive calls may be mutually recursive, § I.4 formalizes this semantics as the least tuple $\boldsymbol{\phi}^{\mathcal{S},\mathcal{F}} = (\phi_1^{\mathcal{S},\mathcal{F}}, \ldots, \phi_m^{\mathcal{S},\mathcal{F}})$ satisfying these query rules.

Allowing several named scaffolds is only notation. A single scaffold could take a mode tag as part of its input and branch to the corresponding case; writing $S_1, \ldots, S_m$ simply lets us refer to those cases separately.

The basic recursive model of § 2 is the one-generator, one-scaffold special case: the scaffold implements the rollout in Algorithm 1 on the active context and then applies the stack-update rule in Equation (2). Figure 2 illustrates three representative examples: summarization, discrete diffusion, and prover/verifier recursion. In all three, recursion depth counts nested recursive scaffold calls, not ordinary generator/tool queries or loop iterations inside one run.

## 4.2. Optimality of Recursive Models

We now compare the general scaffold model with the minimal recursive model analyzed in § 3. The question is whether these more general controllers can compute more under the same local-space bound. The answer is no: once local space and recursion depth are fixed, richer controllers give no additional asymptotic power.

**Definition 4** (L-bounded execution). Fix $(\mathcal{S}, \mathcal{F})$ and its induced partial functions $\boldsymbol{\phi}^{\mathcal{S},\mathcal{F}}$. For $r \in \{1, \ldots, m\}$, input $x \in \Sigma^*$, and $L \in \mathbb{N}$, evaluation of $\phi_r^{\mathcal{S},\mathcal{F}}(x)$ is *L-bounded* if every scaffold invocation in the resulting recursive call tree stores at most $L$ symbols locally, including its internal workspace and any query/answer strings it currently holds. This bound is per call frame; the internal computation of generators/tools is not counted.

---

**Algorithm 3** Summarization

**Input:** Input $x$, generator $f$, summarizer $g$, max length $L$.
1: **while** $\neg\mathsf{stop}(x)$ **do**
2:     $y \leftarrow f(x)$                  ▷ *generate*
3:     **if** $|y| \geq L$: $x \leftarrow g(y)$      ▷ *summarize*
4:     **else**: $x \leftarrow y$
5: **return** $x$

---

**Algorithm 4** Discrete Diffusion

**Input:** State $x \in (\Sigma \cup \{\mathsf{mask}\})^n$, denoiser $f$, transition $g$.
1: **while** $\neg\mathsf{stop}(x)$ **do**
2:     $y \leftarrow f(x)$      ▷ *predict mask-free tokens*
3:     $x \leftarrow g(x, y)$      ▷ *new masked sequence*
4: **return** $x$

---

**Algorithm 5** Mutual Recursion: PROVER & VERIFIER

**Input:** Goal $g$, seeds $s_1, \ldots, s_k$, prover $f_p$, verifier $f_v$.
1: **def** PROVER($g$):
2:     **for** $i = 1$ to $k$:
3:         $p \leftarrow f_p(g, s_i)$      ▷ *generate proof*
4:         **if** VERIFIER($g, p$) = `correct`:
5:             **return** `correct`
6:     **return** `wrong`      ▷ *all failed*
7:
8: **def** VERIFIER($g, p$):
9:     $(\mathsf{status}, \mathcal{G}) \leftarrow f_v(g, p)$      ▷ *check proof*
10:     **if** $\mathsf{status} \in \{\mathtt{correct}, \mathtt{wrong}\}$:
11:         **return** status
12:     **if** $\mathsf{status} = \mathtt{incomplete}$:
13:         **return** $\wedge_{g' \in \mathcal{G}}$PROVER($g'$)      ▷ *prove subgoals*

---

*Figure 2.* Examples formalized as recursive agentic systems. **Summarization** (top left) has one scaffold and two generators, a generator $f$ and a summarizer $g$. The scaffold keeps a current sequence $x$, queries $f$ for $y = f(x)$, and if $|y| \geq L$ queries $g$ to compress $y$ before continuing; otherwise it continues from $y$. It makes no recursive call, so $D = 1$. **Discrete diffusion** (bottom left) is also non-recursive: one scaffold maintains a masked sequence $x$, queries the denoiser $f$, and applies the transition rule $g(x, y)$ to reveal, overwrite, or re-mask positions. Iterating this refinement does not create child scaffold runs, so again $D = 1$. **Prover/verifier recursion** (right) uses two scaffolds, $S_{\mathrm{prove}}$ and $S_{\mathrm{verify}}$, and two generators, $f_p$ and $f_v$. The prover uses $f_p$ to propose candidate proofs and calls the verifier; the verifier uses $f_v$ to accept, reject, or identify missing subgoals. If a proof is incomplete, the verifier recursively starts prover runs on the subgoals and returns their conjunction, so the recursive call tree is the proof-decomposition tree.

This is the analogue of the local space bound in Definition 1. Recursion depth controls how many such frames may be nested.

**Unbounded Depth.** The first bound says that any $L$-bounded recursive agentic system can be simulated in time exponential in its per-call local space, relative to its generators.

**Theorem 4** (Upper bounds under $L$-bounded executions (unbounded recursion depth)). *Fix any function $L(n) \geq n$. Let $(\mathcal{S}, \mathcal{F})$ be any recursive agentic system. For any index $r \in \{1, \ldots, m\}$, any language decided by $\phi_r^{\mathcal{S}, \mathcal{F}}$ under $L(n)$-bounded execution for input of length $n$ (Definition 4) lies in $\mathsf{DTIME}^{\mathcal{F}}(2^{\mathcal{O}(L(n))})$. Here $\mathsf{DTIME}^{\mathcal{F}}$ denotes the usual relativized deterministic time class (§ I.3), viewing the generator/tool family $\mathcal{F}$ as an oracle family.*

*In particular, if every generator/tool in $\mathcal{F}$ is computable by a deterministic Turing machine in time $2^{\mathcal{O}(L(n))}$ and work space $\mathcal{O}(L(n))$ on all queries of length at most $L(n)$, then the language lies in $\mathsf{TIME}(2^{\mathcal{O}(L(n))})$.*

**Constant Depth.** If the recursion depth is constant, a depth-first simulation stores only a constant number of $L$-bounded calls, giving a space bound.

**Theorem 5** (Upper bounds under $L$-bounded executions (constant recursion depth)). *Fix any function $L(n) \geq n$. Let $(\mathcal{S}, \mathcal{F})$ be any recursive agentic system. For any index $r \in \{1, \ldots, m\}$, if $\phi_r^{\mathcal{S}, \mathcal{F}}$ decides a language under $L(n)$-bounded execution for input of length $n$ (Definition 4)*

*and the recursion stack depth is $D(n) = \mathcal{O}(1)$ throughout evaluation of $\phi_r^{\mathcal{S}, \mathcal{F}}$, then the decided language lies in $\mathsf{DSPACE}^{\mathcal{F}}(\mathcal{O}(L(n)))$.*

*In particular, if every generator/tool in $\mathcal{F}$ is computable by a deterministic Turing machine in time $2^{\mathcal{O}(L(n))}$ and work space $\mathcal{O}(L(n))$ on all queries of length at most $L(n)$, then the language lies in $\mathsf{DSPACE}(\mathcal{O}(L(n)))$.*

See §§ J and K for the proofs.

Together with the lower bounds in § 3, these results show that recursion, rather than the choice of controller, is the source of the gain: richer fixed controllers may be useful in practice, but they do not asymptotically exceed the simple `call`/`return` model under the same local-space and depth bounds.

## 5. Experiments

We validate the effectiveness of recursive models by training base LLMs on tasks requiring long-horizon reasoning. Below we describe the setup (§5.1) and present results.

### 5.1. Experimental Setup

**Training.** We train recursive models via supervised fine-tuning on generated reasoning traces. Each trace is decomposed into ordinary next-token prediction examples over active contexts. In each example, the input is the text visible in the current stack frame, including the prompt and any previous reasoning in that frame, and the target is the next

*Table 1.* Accuracy (%) on SAT instances. Baseline results from Wei et al. (2025). Ours is fine-tuned from Qwen2.5-3B-Instruct.

| Model | Easy | Medium | Hard |
|---|---|---|---|
| *Random Baseline* | *50.0* | *50.0* | *50.0* |
| DeepSeek-Distill-14B | 84.3 | 55.2 | 46.4 |
| LLaMA3.3-70B | 65.1 | 58.1 | 52.9 |
| Qwen3-235B | 88.0 | 64.8 | 51.4 |
| GPT-4o | 69.9 | 55.2 | 48.8 |
| Recursive Model (ours) | 98 | 95 | 64 |

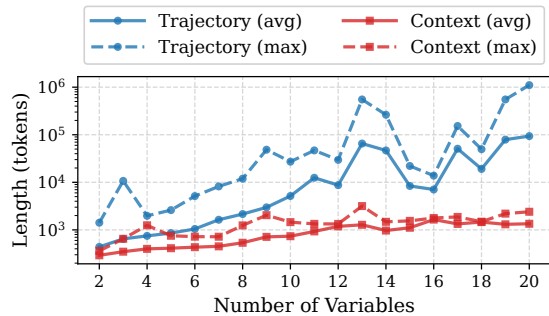

*Figure 3.* Trajectory length vs. active context length.

continuation to generate in that frame, ending at a `call`, a `return`, or the final answer. We apply the standard decoder-only language-modeling loss to these target tokens. Thus, during training, the model predicts continuations from the active context only; suspended parent or child contexts are not included in the conditioning text.

**Dataset.** We evaluate on SAT, a canonical NP-complete problem: given $n$ Boolean variables and $m$ clauses (each a disjunction of literals), determine whether there exists an assignment to the variables such that the entire formula evaluates to true. Worst-case reasoning trajectories grow exponentially, yet with `call` and `return`, each local context stays linear. SAT admits a natural recursive solution: pick an unassigned variable, try setting it to true; if a clause becomes unsatisfied, backtrack and try false. We encode each branch as `call` and each backtrack as `return`. We adopt instances from Wei et al. (2025), converted to natural language puzzles, and generate reasoning traces in our recursive format. Details appear in § B.1.

**Implementation.** We make two practical adaptations. First, upon `return`, we preserve the subtask description and answer in the parent context so the parent knows what was asked and solved. Second, we prepend the root problem to every recursive context so all subtasks retain access to the global objective. See § B.3 for details.

### 5.2. Results

**Accuracy.** We fine-tune Qwen2.5-3B-Instruct with our recursive framework (see §§ B.1 and B.2 for data splits and training details) and compare against frontier LLMs with standard prompting, including GPT-4o, LLaMA3.3-70B, and Qwen3-235B. As Table 1 shows, these larger models struggle on SAT, with accuracy dropping as difficulty increases. In contrast, our recursive model, despite being orders of magnitude smaller, achieves 98% on easy and 95% on medium instances, substantially outperforming baselines. Notably, we train on easy and medium instances, yet the model still reaches 64% on hard instances, suggesting that the recursive decomposition strategy learned from simpler problems can transfer to harder ones beyond the training

distribution.

**Context Efficiency.** Recursive models also enjoy context efficiency. Define *trajectory length* as the total tokens generated across all recursive calls, and *active context length* as the maximum context actually used at any step. Figure 3 shows that trajectory length grows rapidly with problem size, while active context length stays bounded. This gap widens as problems become harder, highlighting the benefit of recursive decomposition: the model can explore deep search trees without exceeding its context limit.

## 6. Discussion

### 6.1. Inference Efficiency

Recursion significantly reduces inference cost by decoupling stack capacity from attention cost. Any single-context model, even those with proper context management strategies such as summarization, must attend to all preceding tokens in the sequence, incurring $\mathcal{O}(|\mathbf{x}_t|)$ FLOPs with KV cache at each step $t$. In contrast, recursive models bound the active context to $|\mathbf{S}_t[-1]| \leq \mathsf{LS}(\mathbf{S}_t)$, therefore requiring only $\mathcal{O}(\mathsf{LS}(\mathbf{S}_t))$ FLOPs per token. This is a $\mathsf{GS}(\mathbf{S}_t)/\mathsf{LS}(\mathbf{S}_t)$ times speedup over the baseline that works on a single sequence, and larger speedup compared with standard CoT that does not manage the context at all. To achieve this speedup, we assume in implementation, KV caches of suspended contexts are stored in external storage and restored upon return, avoiding recomputation.

### 6.2. Heterogeneous Model Selection and Tool-Use

The recursive structure of recursive models naturally supports heterogeneous model selection (Ye et al., 2025; Zhang et al., 2025b; Agashe et al., 2025): instead of always calling itself, the model can invoke different models to handle different subtasks, such as larger models for complex reasoning and smaller models for routine operations. This strikes a natural tradeoff between capability and cost, allowing the overall expense and latency to scale with actual task complexity rather than being dominated by the most expensive model in the system.

### 6.3. Error Accumulation

A potential risk of recursive models is error accumulation: mistakes in subtasks may propagate and corrupt the final answer, especially as recursion depth grows. This concern, however, is not unique to recursion: if CoT produces the same long trajectory as recursive models, a single mistake could propagate as well. Moreover, recursive models offer partial mitigation that CoT lacks: upon `return`, the intermediate reasoning within a subtask is discarded, so errors made there do not pollute sibling or parent computations.

## 7. Related Work

**Recursion in Language Modeling.** Some prior work has explored the idea of recursion in language models. However, these approaches are limited in several ways. **First**, many methods only support shallow recursion (depth = 1) or context folding (Sun et al., 2025; Zhang et al., 2025a; Pan et al., 2025), which we prove in Theorem 3 to be no more powerful than summarization-based single-context models. The concurrent work of Zhang et al. (2025a) focuses on decomposing long inputs, whereas our work studies recursive organization of the reasoning process and proves why recursion depth is the key resource. **Second**, many rely on prompting frozen models to follow recursive patterns (Schroeder et al., 2025; Prasad et al., 2024; Zhang et al., 2025c). **Third**, prior work often targets specific scenarios: arithmetic with fixed recursive patterns (Lee & Kim, 2023), rigid Planner-Executor architectures (Prasad et al., 2024; Zhang et al., 2025c), or context extension via input chunking (Zhang et al., 2025a). This paper provides a general formalization of recursive models, both in its simplest form and generalized form in agentic systems. Our theoretical analysis highlights the critical role of recursion depth: constant-depth recursion offers no advantage over single-context models, whereas unbounded depth unlocks exponentially greater computational power.

**Agentic Systems and Context Management.** LLM-based agentic systems (see Gao et al. (2025); Wang et al. (2024) and references therein) provide a natural setting for recursion: they decompose tasks into modular subtasks handled by agents or tools, often in separate contexts. A related line of work studies how to keep long computations within a bounded context, including *summarization* that compresses context into compact representations (Yang et al., 2025b; Yu et al., 2025; Zhou et al., 2025; Yan et al., 2025; Wu et al., 2025), and *memory augmentation* that maintains external storage for retrieval (Packer et al., 2024; Chhikara et al., 2025; Suzgun et al., 2025; Xu et al., 2025). These approaches address the same pressure from long contexts, but at different levels: agentic systems provide modular control, while context-management methods compress or retrieve information inside a run. Formal analysis remains limited;

notable exceptions are Yang et al. (2025b;c), which focus on summarization and diffusion models respectively.

**Recursion in Classical Computation Theory.** Although the idea that recursion depth and local space are fundamental computational resources has classical roots (Savitch, 1977; Ginsburg et al., 1967; Aho, 1969; Engelfriet, 1991; Savitch, 1970), our work introduces recursion as an explicit design principle for Transformer-based reasoning and proves that constant-depth Transformers can realize the per-step logic at each recursion level (see § A for detailed discussion). More broadly, our results suggest that scaling LLM reasoning need not rely solely on extending context length: a lightweight recursive scaffold that requires no architectural changes can leverage bounded context exponentially more efficiently. Just as recursion transformed programming from flat instruction sequences to modular, composable programs, it may similarly transform LLM reasoning from monolithic chain-of-thought into structured, hierarchical computation.

## 8. Conclusion

We identify recursion as a core principle for overcoming context constraints and propose recursive models as a minimal yet powerful realization. We show that recursion exponentially reduces the required context length compared to single-context approaches, and this power is optimal among all recursive agentic systems. Experiments on SAT validate that even a small model trained with recursive reasoning can significantly outperform frontier LLMs.

## Impact Statement

This paper presents a theoretical understanding of recursive models and suggests an approach to enhance the long-horizon reasoning capabilities of language models. We do not foresee any direct negative societal impact from this work, unless AI systems are employed for unethical purposes, which is a general concern applicable to all advances in machine learning.

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

# A. Recursion in Classical Computation Theory

The idea that recursion depth and local space are fundamental computational resources has deep roots in classical theory. Most directly related to our work, Savitch (1977) formally extended Turing machines with recursive subroutine calls—each call receives its own workspace and returns a result to the caller, mirroring the call/return and context-stack mechanism of our recursive models. Savitch studied the time and storage overhead of recursion, showing that $t$ steps of a recursive TM can be simulated in $O(t)$ steps on a multitape TM, and used this framework to re-derive the $\mathsf{NSPACE}(S) \subseteq \mathsf{DSPACE}(S^2)$ result of Savitch's theorem (Savitch, 1970)—whose proof is itself a recursive subroutine with bounded stack depth, where recursion depth times per-level workspace yields the total space upper bound, foreshadowing our local-vs-global space decomposition. The key difference is that Savitch's recursive TM reads one tape cell per step ($O(1)$ communication), and therefore already captures $\mathsf{SPACE}(S)$ without needing deep recursion. Our recursive model replaces the TM head with a bounded-context Transformer that attends to all $S(n)$ tokens per step; it is this architectural constraint that makes deep recursion necessary to recover the same computational power. Stack automata (Ginsburg et al., 1967) extend pushdown automata by allowing the head to read within the stack, and nested stack automata (Aho, 1969) further allow the creation and destruction of substacks, yielding a stack-of-stacks mechanism reminiscent of our context stack. Engelfriet (1991) studied iterated (higher-order) pushdown storages and established an iterated-exponential hierarchy in computational power as the storage order increases—a phenomenon consistent with our Theorem 1 and Theorem 3. The alternation theorem (Chandra et al., 1981), which we directly use in our proofs, connects alternating computation to space complexity via a recursive evaluation of configuration games.

Our contribution relative to this classical line of work is twofold. First, we introduce recursion as an explicit design principle for Transformer-based reasoning, formalizing how bounded-context language models can overcome their attention bottleneck through recursive self-invocation. Second, we prove *Transformer realizability*: a fixed constant-depth, constant-size Transformer with $\mathcal{O}(\log S(n))$ precision can implement the per-step logic at each recursion level, serving as the transition function of a recursive machine. This bridges the classical recursion-theoretic framework with the concrete capabilities of modern neural architectures.

# B. Experimental Setup

### B.1. Data Generation

We directly use the SAT instances from Wei et al. (2025), which are Boolean formulas in conjunctive normal form (CNF). Each instance is converted to a natural language puzzle where variables map to real-world entities and clauses become narrative constraints. The dataset contains instances of varying difficulty based on the number of clauses: easy (4–19 clauses), medium (20–30 clauses), and hard (31–50 clauses).

For each instance, we generate a recursive reasoning trace by running the DPLL algorithm. At each step, the algorithm picks an unassigned variable and tries assigning it to True. After each assignment, we check for conflicts: either a clause becomes empty (directly violated), or unit clauses force the same variable to both True and False. If a conflict is detected, the algorithm backtracks and tries False. We emit `<call>` when branching and `<return>` when returning. These traces are used for supervised fine-tuning.

For training, we select only easy and medium instances with at most 15 variables. For evaluation, we randomly sample 100 held-out instances from each difficulty level (easy, medium, hard) without any filtering.

### B.2. Training Configuration

We fine-tune from Qwen2.5-3B-Instruct (Yang et al., 2025a), a decoder-only Transformer with 3 billion parameters. We use the AdamW optimizer with a learning rate of $1 \times 10^{-5}$ and cosine decay schedule. The batch size is 16 with gradient checkpointing enabled. We train for 10 epochs with a maximum context length of 4096 tokens (left truncation for sequences exceeding this limit). Training is conducted on $2\times$ NVIDIA H200 GPUs and takes approximately 8 hours.

### B.3. Implementation

When `<call>` is generated, only the reasoning before the tag is preserved in the parent context; the tag content becomes the child's `current_task`. When `<return>` is generated, the subtask description and return value are appended to the parent context in the format "*subtask_desc*. The answer is: *result*." while the intermediate reasoning within the subtask is

discarded. Each context is wrapped with the following template:

```
[Instructions]
Solve problems recursively. Use <call> </call> to decompose
the problem and <return> </return> to return the answer.

[Root Problem]
{root_problem}

[Current Task]
{current_task}
```

The `root_problem` field contains the original problem statement and remains constant across all recursion levels. The `current_task` field contains the subtask description for the current level (equal to `root_problem` at the top level).

### B.4. Sample Problems

Each training sample consists of three parts: `user` (the prompt), `assistant_prefix` (previous reasoning to continue from), and `assistant_content` (the target to predict). Below we show two representative turns.

**Example 1: `<call>` at root level.** The model analyzes clauses and branches on a variable:

```
# user
[Instructions]
Solve problems recursively. Use <call> </call> to decompose the problem and <return> </
    return> to return the answer.

[Root Problem]
Five scientists--Alice, Bob, Carol, Dave, and Eve--are considering whether to join a new
    research project. Each scientist makes an independent decision about their
    participation. They may choose to join or not join the project regardless of others'
    choices.

Conditions:
1. Either Alice joins the project or Carol does not join.
2. Either Alice does not join or Carol joins.
3. Either Alice joins or Carol joins.
4. Either Alice does not join or Carol does not join.
5. Either Carol does not join or Eve does not join.

Is there a way to assign decisions so all these conditions are satisfied?

[Current Task]
Is there a way to assign decisions so all these conditions are satisfied?

# assistant_prefix
(empty)

# assistant_content
Condition 1:
  Clause: (Alice v ~Carol)
  (no simplification needed)
  -> (not unit)
Condition 2:
  Clause: (~Alice v Carol)
  (no simplification needed)
  -> (not unit)
Condition 3:
  Clause: (Alice v Carol)
  (no simplification needed)
  -> (not unit)
Condition 4:
  Clause: (~Alice v ~Carol)
  (no simplification needed)
  -> (not unit)
```

```
Condition 5:
  Clause: (~Carol v ~Eve)
  (no simplification needed)
  -> (not unit)
No unit clause found. Unassigned: [Alice, Carol, Eve]
Try Alice = True
<call>Alice=True</call>
```

**Example 2: `<return>` at leaf node.** The model detects a contradiction and returns:

```
# user
[Instructions]
Solve problems recursively. Use <call> </call> to decompose the problem and <return> </
    return> to return the answer.

[Root Problem]
Five scientists--Alice, Bob, Carol, Dave, and Eve--are considering whether to join a new
    research project. Each scientist makes an independent decision about their
    participation. They may choose to join or not join the project regardless of others'
    choices.

Conditions:
1. Either Alice joins the project or Carol does not join.
2. Either Alice does not join or Carol joins.
3. Either Alice joins or Carol joins.
4. Either Alice does not join or Carol does not join.
5. Either Carol does not join or Eve does not join.

Is there a way to assign decisions so all these conditions are satisfied?

[Current Task]
Alice=True, Carol=True

# assistant_prefix
(empty)

# assistant_content
Given: Alice=True, Carol=True
Condition 1:
  Clause: (Alice v ~Carol)
  -> satisfied
Condition 2:
  Clause: (~Alice v Carol)
  -> satisfied
Condition 3:
  Clause: (Alice v Carol)
  -> satisfied
Condition 4:
  Clause: (~Alice v ~Carol)
  Simplify as: () -> CONFLICT
Contradiction!
<return>No</return>
```

**Example 3: `<call>` with prefix (backtracking).** After trying Alice=True and receiving "No", the model backtracks and tries Alice=False:

```
# user
[Instructions]
Solve problems recursively. Use <call> </call> to decompose the problem and <return> </
    return> to return the answer.

[Root Problem]
Five scientists--Alice, Bob, Carol, Dave, and Eve--are considering whether to join a new
    research project. Each scientist makes an independent decision about their
```

```
    participation. They may choose to join or not join the project regardless of others'
    choices.

Conditions:
1. Either Alice joins the project or Carol does not join.
2. Either Alice does not join or Carol joins.
3. Either Alice joins or Carol joins.
4. Either Alice does not join or Carol does not join.
5. Either Carol does not join or Eve does not join.

Is there a way to assign decisions so all these conditions are satisfied?

[Current Task]
Is there a way to assign decisions so all these conditions are satisfied?

# assistant_prefix
Condition 1:
  Clause: (Alice v ~Carol)
  (no simplification needed)
  -> (not unit)
Condition 2:
  Clause: (~Alice v Carol)
  (no simplification needed)
  -> (not unit)
Condition 3:
  Clause: (Alice v Carol)
  (no simplification needed)
  -> (not unit)
Condition 4:
  Clause: (~Alice v ~Carol)
  (no simplification needed)
  -> (not unit)
Condition 5:
  Clause: (~Carol v ~Eve)
  (no simplification needed)
  -> (not unit)
No unit clause found. Unassigned: [Alice, Carol, Eve]
Try Alice = True
Alice=True. The answer is: No.

# assistant_content
Try Alice = False
<call>Alice=False</call>
```

## C. Transformer Architecture

We define the decoder-only Transformer architecture used throughout this paper. Let $\Sigma$ be a finite vocabulary and $d$ be the hidden dimension.

**Token and Positional Embeddings.** A *token embedding* $\mathsf{TE} : \Sigma \to \mathbb{R}^d$ maps each token to a $d$-dimensional vector. A *positional embedding* $\mathsf{PE} : \mathbb{N}^+ \to \mathbb{R}^d$ encodes position information. For an input sequence $(x_1, \ldots, x_n) \in \Sigma^n$, the initial embedding at position $i$ is $h_i^{(0)} = \mathsf{TE}(x_i) + \mathsf{PE}(i)$.

**Attention.** For query, key, and value vectors $(q, k_j, v_j)_{j=1}^n$ where $q, k_j \in \mathbb{R}^{d_k}$ and $v_j \in \mathbb{R}^{d_v}$, the attention output with temperature $\beta > 0$ is:

$$\mathsf{Attn}_\beta(q, \{k_j, v_j\}_{j=1}^n) = \sum_{j=1}^n \alpha_j v_j, \quad \text{where } \alpha = \mathrm{softmax}_\beta\left((q \cdot k_j)_{j=1}^n\right), \tag{13}$$

and $[\mathrm{softmax}_\beta(z)]_i = \exp(z_i/\beta) / \sum_j \exp(z_j/\beta)$.

**Average-Hard Attention (AHA).** Taking the zero-temperature limit $\beta \to 0$ yields average-hard attention (Merrill et al., 2022), which uniformly averages over the maximum-scoring positions:

$$\mathsf{AHA}(q, \{k_j, v_j\}_{j=1}^n) = \frac{1}{|A|} \sum_{j \in A} v_j, \quad \text{where } A = \arg\max_{j \in [n]} \langle q, k_j \rangle. \tag{14}$$

AHA involves only comparisons and uniform averaging, which can be computed exactly in finite precision. All theoretical results in this paper use AHA.

**Multi-Head Self-Attention.** A *multi-head self-attention* layer with $H$ heads is parametrized by projection matrices $W_Q^h, W_K^h, W_V^h \in \mathbb{R}^{d_k \times d}$ and $W_O^h \in \mathbb{R}^{d \times d_k}$ for $h \in [H]$. For embeddings $(h_1, \ldots, h_n)$, the output at position $n$ is:

$$\mathsf{MHA}(h_1, \ldots, h_n) = \sum_{h=1}^{H} W_O^h \cdot \mathsf{AHA}\left(W_Q^h h_n, \{W_K^h h_j, W_V^h h_j\}_{j=1}^n\right). \tag{15}$$

For decoder-only (causal) Transformers, position $n$ attends only to positions $j \leq n$.

**Feed-Forward Layer.** A *feed-forward* layer with width $d_{\mathsf{ff}}$ and activation $\sigma$ is defined as:

$$\mathsf{FF}(h) = W_2 \cdot \sigma(W_1 \cdot h + b_1) + b_2, \tag{16}$$

where $W_1 \in \mathbb{R}^{d_{\mathsf{ff}} \times d}$, $W_2 \in \mathbb{R}^{d \times d_{\mathsf{ff}}}$, and $b_1, b_2$ are bias terms.

**Transformer Layer.** A single *Transformer layer* combines multi-head attention and feed-forward with residual connections:

$$\mathsf{TF}(h_1, \ldots, h_n) = \mathsf{FF}(\tilde{h}_n) + \tilde{h}_n, \quad \text{where } \tilde{h}_n = \mathsf{MHA}(h_1, \ldots, h_n) + h_n. \tag{17}$$

**Next-Token Predictor.** An $L$-layer decoder-only Transformer defines a next-token predictor $f_\theta : \Sigma^* \to \Sigma$ as:

$$f_\theta(x_1, \ldots, x_n) = \arg\max_{x \in \Sigma} \left[W_{\mathsf{dec}} \cdot h_n^{(L)}\right]_x, \tag{18}$$

where $h_n^{(L)}$ is the final-layer embedding at position $n$, computed by stacking $L$ Transformer layers on top of the initial embeddings, and $W_{\mathsf{dec}} \in \mathbb{R}^{|\Sigma| \times d}$ is the decoding matrix.

**Precision.** We say a Transformer has $\mathcal{O}(\log S(n))$ *precision* if all intermediate numerical values (embeddings, attention scores, and feed-forward activations) are rational numbers $p/q$ with $|p|, |q| \leq S(n)^C$ for a universal constant $C$, where $S(n)$ is the local space bound, i.e., the input sequence length to the Transformer. Equivalently, each value is representable in $\mathcal{O}(\log S(n))$ bits, and all arithmetic is exact with no rounding. This precision model is consistent with Yang et al. (2025b): operations such as `seq_sum` over $S(n)$ indicator values produce results bounded by $S(n)$, `seq_max` preserves input magnitudes, and `rightmost_exact_match` concentrates attention on a single position, all within $\mathcal{O}(\log S(n))$-bit exact arithmetic.

# D. Single-Tape Turing Machine

A single-tape Turing machine operates on an infinite tape indexed by $\mathbb{Z}$, where each cell holds a symbol from a finite tape alphabet $\Gamma$. A read/write head moves along the tape, and a finite set of control states governs the machine's behavior. Formally, a Turing machine is a 7-tuple $\mathsf{TM} = (\Gamma, b, Q, q_0, \delta, Q_{\mathsf{acc}}, Q_{\mathsf{rej}})$, where $b \in \Gamma$ is the blank symbol; $q_0 \in Q$ is the initial state; $\delta : (Q \setminus (Q_{\mathsf{acc}} \cup Q_{\mathsf{rej}})) \times \Gamma \to Q \times \Gamma \times \{-1, 0, +1\}$ is the transition function; and $Q_{\mathsf{acc}}, Q_{\mathsf{rej}} \subseteq Q$ are disjoint accepting and rejecting states.

**Execution.** Given input $x \in (\Gamma \setminus \{b\})^n$, the tape is initialized with $x$ in cells $0, \ldots, n-1$ and blanks elsewhere; the head starts at position $0$ in state $q_0$. At each step, the machine reads the symbol $a$ under the head, computes $(q', w, d) = \delta(q, a)$, writes $w$, moves the head by $d \in \{-1, 0, +1\}$, and transitions to state $q'$. The machine halts upon entering $Q_{\mathsf{acc}} \cup Q_{\mathsf{rej}}$, outputting 1 (accept) or 0 (reject) accordingly.

**Normalization.** To ensure configurations are well-defined for all $t \leq T(n)$, we extend $\delta$ to halting states by making them self-loops: for all $q \in Q_{\mathsf{acc}} \cup Q_{\mathsf{rej}}$ and $a \in \Gamma$, define $\delta(q, a) := (q, a, 0)$. This does not change the language decided by TM.

**Complexity Classes.** The *time complexity* $T(\mathsf{TM}, x)$ is the number of steps before halting. The *space complexity* $S(\mathsf{TM}, x)$ is the number of distinct tape cells visited. A Turing machine $\mathsf{TM}$ *decides* a language $L \subseteq \Sigma^*$ if it halts on all inputs and accepts exactly those in $L$. The complexity classes are defined as:

$$\mathsf{TIME}(f(n)) = \{L : \exists\,\mathsf{TM}\text{ deciding } L \text{ with } T(\mathsf{TM}, x) \leq f(|x|) \text{ for all } x\}, \tag{19}$$

$$\mathsf{SPACE}(f(n)) = \{L : \exists\,\mathsf{TM}\text{ deciding } L \text{ with } S(\mathsf{TM}, x) \leq f(|x|) \text{ for all } x\}. \tag{20}$$

We write $\mathsf{TM}(s(n), t(n))$ for the simultaneous space-time class: languages decided by a single deterministic Turing machine whose space and time on every input $x$ are at most $\mathcal{O}(s(|x|))$ and $\mathcal{O}(t(|x|))$, respectively.

# E. Proof of Theorem 1

**Theorem 6** (Deep Recursive Models, Formal)**.** *For any $S(n) \geq n$, recursive models can solve any problem in* $\mathsf{TIME}(2^{\mathcal{O}(S(n))})$ *under local space constraint $\mathcal{O}(S(n))$:*

$$\mathsf{TIME}(2^{\mathcal{O}(S(n))}) \subseteq \mathsf{RM}(\mathcal{O}(S(n)), \infty, \infty). \tag{21}$$

The proof proceeds in two parts: **(1)** we define mutually recursive functions that compute TM configurations and prove their correctness; **(2)** we analyze the resource consumption (local space, recursion depth, and runtime). We further provides a sketch for constructing the Transformer but the detailed implementation is omitted. An alternative proof via Alternating Turing Machines appears in Appendix F, which includes the detailed implementation for the corresponding Transformer.

## E.1. Recursive Construction

Let $\mathsf{TM} = (\Gamma, b, Q, q_0, \delta, Q_{\mathrm{acc}}, Q_{\mathrm{rej}})$ be a single-tape Turing machine. We use time $t$ to denote the number of transitions already executed: $t = 0$ is the initial configuration, and transitioning from $t$ to $t + 1$ executes the $t$-th transition.

**Configuration.** A *configuration* of $\mathsf{TM}$ at time $t$ is a triple $c_t = (q_t, \tau_t, p_t)$ where:

- $q_t \in Q$ is the control state at time $t$;
- $\tau_t : \mathbb{Z} \to \Gamma$ is the *tape contents* at time $t$, mapping each cell index to a symbol, with $\tau_t(i) = b$ (the blank symbol) for all but finitely many $i$;
- $p_t \in \mathbb{Z}$ is the head position at time $t$.

For a tape $\tau$ and position $p$, we write $\tau[p \mapsto w]$ for the tape that agrees with $\tau$ everywhere except at position $p$, where it holds symbol $w$. The initial configuration is $c_0 = (q_0, \tau_0, 0)$ where $\tau_0(i) = x[i]$ for $0 \leq i < n$ and $\tau_0(i) = b$ otherwise.

**Recursive functions.** We define the following mutually recursive functions that compute the components of $c_t$. Let $x \in (\Gamma \setminus \{b\})^n$ be the input.

- $\mathsf{STATE}(x, t) \in Q$: returns the control state $q_t$
- $\mathsf{POS}(x, t) \in \mathbb{Z}$: returns the head position $p_t$
- $\mathsf{CELL}(x, t, p) \in \Gamma$: returns the tape symbol $\tau_t(p)$ at position $p$
- $\mathsf{SYMBOL}(x, t) \in \Gamma$: returns the symbol under the head $\tau_t(p_t)$
- $\mathsf{RUN}(x, t) \in \{0, 1\}$: starting from time $t$, simulate until halting and return accept (1) or reject (0)

**Algorithm.** The following five algorithms present the pseudocode for these mutually recursive functions. The transition function $\delta$ is assumed to be hardcoded into the model parameters. We fix a constant $c > 0$ such that the Turing machine $\mathsf{TM}$ deciding $L$ halts within $T(n) := 2^{c \cdot S(n)}$ steps on all inputs of length $n$.

---
**Algorithm 6** $\mathsf{STATE}(x, t) \to q_t \in Q$

---
1: **if** $t = 0$ **then return** $q_0$
2: $(q', w, d) \leftarrow \delta(\mathsf{STATE}(x, t - 1), \mathsf{SYMBOL}(x, t - 1))$
3: **return** $q'$

---

---

**Algorithm 7** $\text{POS}(x, t) \to p_t \in \mathbb{Z}$

---

1: **if** $t = 0$ **then return** $0$
2: $(q', w, d) \leftarrow \delta(\text{STATE}(x, t-1), \text{SYMBOL}(x, t-1))$
3: **return** $\text{POS}(x, t-1) + d$

---

---

**Algorithm 8** $\text{CELL}(x, t, p) \to \tau_t(p) \in \Gamma$

---

1: **if** $t = 0$ **then return** $x[p]$ if $0 \leq p < |x|$ else $b$
2: $p_{\text{prev}} \leftarrow \text{POS}(x, t-1)$
3: **if** $p \neq p_{\text{prev}}$ **then return** $\text{CELL}(x, t-1, p)$                                                                 ▷ recurse
4: $(q', w, d) \leftarrow \delta(\text{STATE}(x, t-1), \text{SYMBOL}(x, t-1))$
5: **return** $w$                                                                                 ▷ symbol written at $t-1$

---

---

**Algorithm 9** $\text{SYMBOL}(x, t) \to \tau_t(p_t) \in \Gamma$

---

1: **return** $\text{CELL}(x, t, \text{POS}(x, t))$

---

---

**Algorithm 10** $\text{RUN}(x, t) \to \{0, 1\}$

---

1: $q \leftarrow \text{STATE}(x, t)$
2: **if** $q \in Q_{\text{acc}}$ **then return** $1$                                                                           ▷ accept
3: **if** $q \in Q_{\text{rej}}$ **then return** $0$                                                                           ▷ reject
4: **return** $\text{RUN}(x, t+1)$                                                                                     ▷ continue

---

The decision procedure is $\text{DECIDE}(x) := \text{RUN}(x, 0)$. Since $L \in \text{TIME}(2^{\mathcal{O}(S(n))})$, the TM halts within $T = 2^{c \cdot S(n)}$ steps, so RUN terminates and correctly outputs accept/reject. We now show that the recursive semantics faithfully tracks the TM's behavior.

**Lemma 7** (Correctness of Recursive Semantics). *Let* $q_t, p_t, \tau_t$ *denote the true state, head position, and tape contents of* TM *at time $t$. For every input $x$, every $t \geq 0$, and every position $p \in \mathbb{Z}$:*

1. $\text{STATE}(x, t) = q_t$

2. $\text{POS}(x, t) = p_t$

3. $\text{CELL}(x, t, p) = \tau_t(p)$

4. $\text{SYMBOL}(x, t) = \tau_t(p_t)$

*Proof.* By induction on $t$.

*Base case ($t = 0$):* By the TM initialization semantics, $q_0$ is the initial state, $p_0 = 0$, and $\tau_0(p) = x[p]$ for $0 \leq p < n$ and $\tau_0(p) = b$ otherwise. These match the base cases of our recursive functions. For claim (4), $\text{SYMBOL}(x, 0) = \text{CELL}(x, 0, \text{POS}(x, 0)) = \tau_0(0) = \tau_0(p_0)$.

*Inductive step ($t \geq 1$):* Assume the claims hold for time $t - 1$. By definition and the induction hypothesis:

$$\text{SYMBOL}(x, t-1) = \text{CELL}(x, t-1, \text{POS}(x, t-1)) = \tau_{t-1}(p_{t-1}) \tag{22}$$

Let $(q', w, d) := \delta(\text{STATE}(x, t-1), \text{SYMBOL}(x, t-1))$ be the transition output. By the induction hypothesis, $\text{STATE}(x, t-1) = q_{t-1}$, so the transition $\delta(q_{t-1}, \tau_{t-1}(p_{t-1}))$ computed by the algorithm is exactly the transition taken by TM at step $t - 1$. Thus:

• $\text{STATE}(x, t) = q' = q_t$ (the new state from $\delta$)

• $\text{POS}(x, t) = p_{t-1} + d = p_t$ (head moves by $d$)

- $\mathsf{CELL}(x, t, p) = \tau_t(p)$: only cell $p_{t-1}$ changes to $w$; others unchanged
- $\mathsf{SYMBOL}(x, t) = \mathsf{CELL}(x, t, \mathsf{POS}(x, t)) = \tau_t(p_t)$

This completes the induction. $\square$

### E.2. Resource Analysis

We analyze three resources: local space (per-context length), recursion depth (call stack height), and total runtime (number of recursive calls).

**Local Space.** Each recursive frame must store the following data:

- *Input $x$*: length $n$
- *Time parameter $t'$*: $O(\log t)$ bits in binary representation
- *Position parameter $p$ (for $\mathsf{CELL}$)*: since the head moves at most 1 cell per step, $|p| \leq t$, so $|\mathrm{bin}(p)| = O(\log t)$
- *State $q \in Q$, symbol $a \in \Gamma$, move direction $d \in \{-1, 0, +1\}$*: $O(1)$ bits (finite sets)
- *Returned answers from subcalls*: state ($O(1)$), position ($O(\log t)$), symbol ($O(1)$)

Crucially, each context makes only $O(1)$ nested calls before returning. When a callee returns, the call/return mechanism removes its entire context from the stack and appends only the returned value to the caller's context. This prevents accumulation of intermediate results. Thus, each context has length $O(n + \log t)$. In Theorem 6, $t \leq T(n) = 2^{c \cdot S(n)}$, so $\log t \leq c \cdot S(n)$. Since $S(n) \geq n$, the local space bound is $O(n + S(n)) = O(S(n))$. Moreover, the complete rollout before each call or return adds only a constant number of delimiters and a subcall prompt or returned value of length $O(n + \log t)$, so the transient rollout stack also has local space $O(S(n))$.

**Recursion Depth.** For the inner functions ($\mathsf{STATE}, \mathsf{POS}, \mathsf{CELL}, \mathsf{SYMBOL}$): each call with time parameter $t$ recursively invokes only subcalls with parameter $t - 1$. Thus, starting from $t = T(n)$, the recursion depth is $O(T(n))$.

For the outer decision procedure $\mathsf{RUN}$: even without assuming any tail-call optimization, the additional stack height contributed by iterating through time steps $0, 1, \ldots$ is at most $O(T(n))$. Each $\mathsf{RUN}(x, t)$ calls $\mathsf{STATE}(x, t)$, which itself has depth $O(t)$.

Overall, the maximum recursion depth is $O(T(n)) = O(2^{c \cdot S(n)})$.

**Time Complexity (Total Subroutine Invocations).** We measure runtime by the total number of subroutine invocations across all recursive contexts. Since the Transformer $f_\theta$ has constant size and each invocation produces at most $O(S(n))$ tokens, this differs from the total token count by at most an $O(S(n))$ factor.

For each routine $\mathsf{F} \in \{\mathsf{STATE}, \mathsf{POS}, \mathsf{CELL}, \mathsf{SYMBOL}, \mathsf{RUN}\}$, let $\mathcal{T}_\mathsf{F}(t)$ denote the worst-case total number of subroutine invocations triggered by evaluating $\mathsf{F}(x, t)$ (for $\mathsf{CELL}$, we also maximize over $p \in \mathbb{Z}$). From Algorithms 1–5:

$$\mathcal{T}_\mathsf{STATE}(t) \leq \mathcal{T}_\mathsf{STATE}(t-1) + \mathcal{T}_\mathsf{SYMBOL}(t-1) + O(1), \tag{23}$$
$$\mathcal{T}_\mathsf{POS}(t) \leq \mathcal{T}_\mathsf{POS}(t-1) + \mathcal{T}_\mathsf{STATE}(t-1) + \mathcal{T}_\mathsf{SYMBOL}(t-1) + O(1), \tag{24}$$
$$\mathcal{T}_\mathsf{CELL}(t) \leq \mathcal{T}_\mathsf{POS}(t-1) + \max\{\mathcal{T}_\mathsf{CELL}(t-1), \mathcal{T}_\mathsf{STATE}(t-1) + \mathcal{T}_\mathsf{SYMBOL}(t-1)\} + O(1), \tag{25}$$
$$\mathcal{T}_\mathsf{SYMBOL}(t) \leq \mathcal{T}_\mathsf{POS}(t) + \mathcal{T}_\mathsf{CELL}(t) + O(1), \tag{26}$$
$$\mathcal{T}_\mathsf{RUN}(t) \leq \mathcal{T}_\mathsf{STATE}(t) + \mathcal{T}_\mathsf{RUN}(t+1) + O(1). \tag{27}$$

To simplify these coupled recurrences, we define two dominant quantities:

$$V(t) := \max\{\mathcal{T}_\mathsf{STATE}(t), \mathcal{T}_\mathsf{POS}(t)\}, \qquad C(t) := \mathcal{T}_\mathsf{CELL}(t). \tag{28}$$

By equation 26, $\mathcal{T}_\mathsf{SYMBOL}(t) \leq V(t) + C(t) + O(1)$. Substituting into equation 23–equation 25 yields:

$$V(t) \leq 3V(t-1) + C(t-1) + O(1),$$
$$C(t) \leq 3V(t-1) + C(t-1) + O(1).$$

Therefore,

$$\begin{pmatrix} V(t) \\ C(t) \end{pmatrix} \preceq \begin{pmatrix} 3 & 1 \\ 3 & 1 \end{pmatrix} \begin{pmatrix} V(t-1) \\ C(t-1) \end{pmatrix} + O(1), \tag{29}$$

whose spectral radius is 4. Hence $V(t), C(t) = O(4^t)$.

Finally, if the simulated Turing machine halts within $T(n)$ steps, then $\mathsf{RUN}(x, 0)$ performs at most $T(n)$ iterations, each invoking $\mathsf{STATE}(x, t)$ once. Using equation 27:

$$\mathcal{T}_{\mathsf{RUN}}(0) \leq \sum_{t=0}^{T(n)} \mathcal{T}_{\mathsf{STATE}}(t) = O(V(T(n))) = O(4^{T(n)}). \tag{30}$$

For $T(n) = 2^{c \cdot S(n)}$, this becomes $4^{T(n)} = 2^{2T(n)} = 2^{2^{\Theta(S(n))}}$, i.e., double exponential in $S(n)$. Since each invocation produces at most $O(S(n))$ tokens, the total generated tokens remain $2^{2^{\Theta(S(n))}}$.

This double-exponential runtime does *not* affect the $\mathsf{RM}(\cdot, \cdot)$ membership statement, which constrains only local space and recursion depth. To reduce runtime to $2^{\mathcal{O}(S(n))}$, one can augment the simulation with memoization: caching results of $\mathsf{STATE}(x, t')$ in external storage ensures each subproblem is computed only once.

### E.3. Transformer Construction

We sketch how a Transformer can implement the recursive simulation described above. The key insight is that each step of Algorithms 1–5 involves only: (i) parsing a bounded-length prefix to identify the function and arguments, (ii) counting delimiters to determine the current phase, (iii) performing constant-size table lookups ($\delta$, $Q_{\mathrm{acc}}$, $Q_{\mathrm{rej}}$), and (iv) emitting tokens for calls/returns.

#### E.3.1. SETUP

**Token Vocabulary.** We define the following special tokens:

- *Function tokens*: $\langle\mathsf{STATE}\rangle$, $\langle\mathsf{POS}\rangle$, $\langle\mathsf{CELL}\rangle$, $\langle\mathsf{SYMBOL}\rangle$, $\langle\mathsf{RUN}\rangle$ indicate which recursive function is being invoked.

- *Control tokens*: $\langle\mathtt{call}\rangle$, $\langle\mathtt{/call}\rangle$, $\langle\mathtt{return}\rangle$, $\langle\mathtt{/return}\rangle$ mark the boundaries of recursive calls and returns.

- *Separator tokens*: $\langle\mathtt{sep}\rangle$ separates arguments within a call; $[\mathtt{SEP}]$ is an internal delimiter that separates cached intermediate results within a context.

- *Data tokens*: Tokens from $\Gamma$ (tape alphabet), $Q$ (states), and binary digits $\{0, 1\}$ for encoding integers.

**Context Format.** Each function-call frame is a single sequence (context) whose prefix contains the input arguments, and whose suffix progressively caches intermediate results from subcalls. A typical context has the form:

$$\langle F \rangle \ \langle\mathtt{sep}\rangle \ \mathrm{arg}_1 \ \langle\mathtt{sep}\rangle \ \mathrm{arg}_2 \ \big( \ \langle\mathtt{sep}\rangle \ \mathrm{arg}_3 \ \big) \ [\mathtt{SEP}] \ z_1 \ [\mathtt{SEP}] \ z_2 \ \cdots \tag{31}$$

where $\langle F \rangle$ is the function token (e.g., $\langle\mathsf{STATE}\rangle$), and each $z_i$ is either a returned value from a recursive call or an internally produced constant-size token. Recursive calls are wrapped as $\langle\mathtt{call}\rangle\langle F\rangle\langle\mathtt{sep}\rangle\cdots\langle\mathtt{/call}\rangle$. Returns are encoded as $\langle\mathtt{return}\rangle[\mathtt{SEP}] \parallel \mathbf{v}\langle\mathtt{/return}\rangle$—note that the payload begins with $[\mathtt{SEP}]$. Thus, when a subcall finishes, the caller receives the payload $[\mathtt{SEP}] \parallel \mathbf{v}$ appended to its context, so each completed subcall contributes exactly one $[\mathtt{SEP}]$ delimiter to the caller's phase cache.

**Transformer Behavior.** The Transformer $f_\theta$ decides what to do next by inspecting only: (i) which function token $\langle F \rangle$ begins the context, (ii) whether the time argument is zero (via a bit-scan), and (iii) how many $[\mathtt{SEP}]$ delimiters have already appeared (the "phase"). This is a standard finite-phase construction: each function needs only a constant number of phases to implement the corresponding algorithmic step. Specifically, $f_\theta$ executes the logic specified in Algorithms 1–5:

1. **Base case**: If $t = 0$ (detected by checking if $\mathrm{bin}(t)$ is all zeros), output $\langle\mathtt{return}\rangle[\mathtt{SEP}] \parallel \mathbf{v}_0\langle\mathtt{/return}\rangle$ where $\mathbf{v}_0$ is the base case value ($q_0$, $0$, $x[p]$ or $b$, depending on the function).

2. **Recursive case**: If $t > 0$, the Transformer performs the following operations depending on the function token:

- ⟨**STATE**⟩: (i) compute $t - 1$ via binary decrement; (ii) call ⟨STATE⟩ and ⟨SYMBOL⟩ with $(x, t - 1)$ to obtain $q_{t-1}$ and $a_{t-1}$; (iii) compute $\delta(q_{t-1}, a_{t-1})$ via lookup table to get $(q', w, d)$; (iv) return $q'$.

- ⟨**POS**⟩: (i) compute $t - 1$; (ii) call ⟨STATE⟩ and ⟨SYMBOL⟩ with $(x, t - 1)$; (iii) compute $\delta$ to get $d$; (iv) call ⟨POS⟩ with $(x, t - 1)$ to get $p_{t-1}$; (v) compute $p_{t-1} + d$ via binary addition; (vi) return $p_t$.

- ⟨**CELL**⟩: (i) compute $t - 1$; (ii) call ⟨POS⟩ with $(x, t - 1)$ to get $p_{t-1}$; (iii) compare $p$ with $p_{t-1}$: if $p \neq p_{t-1}$, recurse by calling ⟨CELL⟩ with $(x, t - 1, p)$; otherwise (iv) call ⟨STATE⟩ and ⟨SYMBOL⟩ with $(x, t - 1)$, compute $\delta$ to get $w$, and return $w$.

- ⟨**SYMBOL**⟩: call ⟨POS⟩ with $(x, t)$ to get $p_t$, then call ⟨CELL⟩ with $(x, t, p_t)$ and return the result.

- ⟨**RUN**⟩: (i) call ⟨STATE⟩ with $(x, t)$ to get $q_t$; (ii) check if $q_t \in Q_{\mathrm{acc}} \cup Q_{\mathrm{rej}}$: if $q_t \in Q_{\mathrm{acc}}$, return 1; if $q_t \in Q_{\mathrm{rej}}$, return 0; otherwise (iii) compute $t + 1$ via binary increment and call ⟨RUN⟩ with $(x, t + 1)$.

3. **Return processing**: When a ⟨/return⟩ token is encountered, the stack-transition rule Step pops the current frame and appends the payload [SEP] ∥ **v** to the parent context, automatically incrementing the parent's phase count.

**Transformer construction.** It remains to verify that the next-token policy described above is implementable by a fixed constant-depth, constant-size Transformer with $\mathcal{O}(\log S(n))$ precision. The recursive functions STATE, POS, CELL, SYMBOL, and RUN reduce to the following primitive operations:

(a) Parsing the context to identify the function token ⟨F⟩ and extract arguments;

(b) Phase counting via seq_sum: counting the number of [SEP] delimiters to determine the current computation phase;

(c) Binary arithmetic: increment $(t \mapsto t + 1)$ and decrement $(t \mapsto t - 1)$ of the time parameter, and position updates $(p \pm d)$, using seq_max for bit-scans;

(d) Cache retrieval via rightmost_exact_match: retrieving previously computed values from the context;

(e) Finite lookups of $\delta, Q_{\mathrm{acc}}, Q_{\mathrm{rej}}$ (hard-coded into parameters).

All primitive operations (a)–(e) above are already established in Appendix G of Yang et al. (2025b); our construction differs only in the choice of special tokens and parsing format. We refer readers to that paper for the detailed Transformer implementation. A complete construction using an alternative approach (via Alternating Turing Machines) appears in Appendix F.

# F. Proof of Theorem 1 via Alternating Turing Machine

This section gives an alternative proof of Theorem 1. The proof follows the classical characterization $\mathsf{TIME}(2^{O(S(n))}) = \mathsf{ASPACE}(O(S(n)))$ (Chandra–Kozen–Stockmeyer) and then realizes the resulting AND/OR computation using the call/return recursion mechanism, with the per-step logic implemented by a constant-depth Transformer via Full-Access Sequence Processing (FASP) (Yang et al., 2025b).

## F.1. Alternating Turing Machines and ASPACE

An *alternating Turing machine* (ATM) is a nondeterministic Turing machine $A = (\Gamma, b, Q, q_0, \Delta, Q_{\mathrm{acc}}, Q_{\mathrm{rej}})$ whose non-halting states are partitioned into *existential* and *universal* states: $Q \setminus (Q_{\mathrm{acc}} \cup Q_{\mathrm{rej}}) = Q_\exists \,\dot\cup\, Q_\forall$. The transition relation is a finite set

$$\Delta \subseteq (Q \setminus (Q_{\mathrm{acc}} \cup Q_{\mathrm{rej}})) \times \Gamma \times Q \times \Gamma \times \{-1, 0, +1\}. \tag{32}$$

Each tuple $(q, a, q', a', d) \in \Delta$ specifies: in state $q$ reading symbol $a$, the machine may transition to state $q'$, write $a'$ on the current cell, and move the head by $d \in \{-1, 0, +1\}$. For a configuration $c = (q, \tau, p)$ (state $q$, tape contents $\tau : \mathbb{Z} \to \Gamma$, head position $p$), the set of successor configurations is

$$\mathrm{Succ}(c) := \{(q', \tau[p \mapsto a'], p + d) : (q, \tau(p), q', a', d) \in \Delta\}, \tag{33}$$

where $\tau[p \mapsto a']$ denotes the tape with symbol at position $p$ updated to $a'$. Since we assume exactly two successors, we index them as $\mathrm{Succ}_0(c)$ and $\mathrm{Succ}_1(c)$. For $i \in \{0, 1\}$, let $\delta_i(c) := (q_i', w_i, d_i)$ denote the $i$-th applicable transition tuple (i.e., $(q, \tau(p), q_i', w_i, d_i) \in \Delta$), so that $\mathrm{Succ}_i(c) = (q_i', \tau[p \mapsto w_i], p + d_i)$.

**Acceptance Semantics.** Fix an input $x$ and let $c_{\text{start}}(x)$ be the start configuration. Assuming $A$ is a *decider* (every branch halts), the acceptance value $\text{Win}(c) \in \{0, 1\}$ is defined recursively over the computation tree: if $c$ halts in $Q_{\text{acc}}$ then $\text{Win}(c) = 1$; if in $Q_{\text{rej}}$ then $\text{Win}(c) = 0$; if $c$ is non-halting with state in $Q_\exists$, then $\text{Win}(c) = \bigvee_{c' \in \text{Succ}(c)} \text{Win}(c')$; if in $Q_\forall$, then $\text{Win}(c) = \bigwedge_{c' \in \text{Succ}(c)} \text{Win}(c')$. The machine accepts $x$ iff $\text{Win}(c_{\text{start}}(x)) = 1$.

**Alternating Space.** The class $\text{ASPACE}(S(n))$ consists of languages decidable by an ATM that visits at most $O(S(n))$ tape cells along every branch.

**Lemma 8** (Chandra–Kozen–Stockmeyer characterization)**.** *For any space-constructible $S(n) \geq n$,*

$$\text{TIME}(2^{O(S(n))}) = \text{ASPACE}(O(S(n))). \tag{34}$$

*Proof.* This is the classical *alternation theorem* (Chandra et al., 1981); see also standard textbook treatments (Arora & Barak, 2009). □

### F.2. Recursive Construction

Fix a space-constructible $S(n) \geq n$ and a language $L \in \text{TIME}(2^{O(S(n))})$. By Lemma 8, there exists an ATM $A$ deciding $L$ in space $O(S(n))$. Deciding $x$ reduces to evaluating $\text{Win}(c_{\text{start}}(x))$. Since $A$ is fixed, we assume w.l.o.g. that every non-halting configuration has *exactly two* successors (by padding missing successors with reject for existential states and accept for universal states, and converting bounded fanout to binary). We denote the two successors by $\text{Succ}_0(c)$ and $\text{Succ}_1(c)$.

**Configuration.** A *configuration* of $A$ is a triple $c = (q, \tau, p)$ where $q \in Q$ is the control state, $\tau : \mathbb{Z} \to \Gamma$ is the tape contents, and $p \in \mathbb{Z}$ is the head position. Since $A$ uses $O(S(n))$ space, each reachable configuration can be encoded as a token sequence $\text{Embed}(c)$ of length $O(S(n))$; the precise encoding is described in § F.3.

**Recursive functions.** We define the following functions for evaluating configurations:

- $\text{STEP}(c, i) \in \{\text{configurations}\}$: returns the $i$-th successor $\text{Succ}_i(c)$ for $i \in \{0, 1\}$

- $\text{HALTING}(c) \in \{0, 1, \bot\}$: returns 1 if $c \in Q_{\text{acc}}$, 0 if $c \in Q_{\text{rej}}$, $\bot$ otherwise

- $\text{TYPE}(c) \in \{\exists, \forall\}$: returns the alternation type of non-halting configuration $c$

- $\text{COMB}(c, b_0, b_1) \in \{0, 1\}$: returns $b_0 \vee b_1$ if $\text{TYPE}(c) = \exists$, else $b_0 \wedge b_1$

- $\text{EVAL}(c) \in \{0, 1\}$: evaluates $\text{Win}(c)$ recursively

**Algorithm.** The following algorithm presents the pseudocode for EVAL:

---
**Algorithm 11** $\text{EVAL}(c) \to b \in \{0, 1\}$
---
1: **if** $\text{HALTING}(c) = 1$ **then return** 1 ▷ accept
2: **if** $\text{HALTING}(c) = 0$ **then return** 0 ▷ reject
3: $b_0 \leftarrow \text{EVAL}(\text{STEP}(c, 0))$ ▷ evaluate first successor
4: $b_1 \leftarrow \text{EVAL}(\text{STEP}(c, 1))$ ▷ evaluate second successor
5: **return** $\text{COMB}(c, b_0, b_1)$ ▷ AND/OR combination
---

**Correctness.** By structural induction on the computation tree:

- *Base case:* If $c$ is halting, $\text{EVAL}(c)$ returns 1 iff $c \in Q_{\text{acc}}$, which equals $\text{Win}(c)$ by definition.

- *Inductive step:* If $c$ is non-halting, by IH, $b_i = \text{EVAL}(\text{STEP}(c, i)) = \text{Win}(\text{Succ}_i(c))$ for $i \in \{0, 1\}$. Then $\text{COMB}(c, b_0, b_1)$ computes the correct AND/OR combination based on $\text{TYPE}(c)$, matching the definition of $\text{Win}(c)$.

Thus $\text{EVAL}(c_{\text{start}}(x)) = \text{Win}(c_{\text{start}}(x))$, correctly deciding whether $A$ accepts $x$.

**Resource analysis.** Each recursive frame stores the configuration encoding $\mathsf{Embed}(c)$ ($O(S(n))$ tokens), the returned bits $b_0, b_1$ ($O(1)$ bits), and call/return delimiters ($O(1)$ tokens), yielding local space $O(S(n))$ per context. The generated call payloads $\mathsf{Embed}(c_i)$ also have length $O(S(n))$, and return payloads have length $O(1)$, so the transient rollout stacks satisfy the same local-space bound. For recursion depth, an ATM using $O(S(n))$ space has at most $2^{O(S(n))}$ distinct configurations (finite control $\times$ head position $\times$ tape contents). Because $A$ is a decider, the configuration graph is acyclic—a cycle would induce an infinite branch. Hence the maximum recursion depth is bounded by the number of reachable configurations: $2^{O(S(n))}$.

## F.3. Preliminaries and Setup

To implement the recursive evaluation with a Transformer, we first introduce how to represent Turing machine configurations as token sequences that the Transformer can process.

**Update tokens.** We encode configurations using *update tokens*. Let $\Sigma_{\mathrm{upd}} := Q \times \Gamma \times \{-1, 0, +1\}$ be the set of update tokens, where each token $(q', w, d)$ represents: "write $w$ at the current head cell, move by $d$, and set state to $q'$".

**Update operator.** For a configuration $c = (q, \tau, p)$, define the *update operator* $\mathsf{Update}(c, (q', w, d)) := (q', \tau[p \mapsto w], p + d)$, and extend it to sequences by $\mathsf{Update}(c, x_{1:k}) := \mathsf{Update}(\mathsf{Update}(c, x_{1:k-1}), x_k)$. Let $c_{\mathsf{blank}} := (q_0, b^{\mathbb{Z}}, 0)$ denote the blank configuration (initial state, all-blank tape, head at origin).

**Translational equivalence.** Two configurations $c_1 = (q, \tau_1, p_1)$ and $c_2 = (q, \tau_2, p_2)$ are *translationally equivalent*, written $c_1 \sim c_2$, if there exists $k \in \mathbb{Z}$ such that $\tau_1(i) = \tau_2(i - k)$ for all $i$ and $p_1 = p_2 + k$. Intuitively, they differ only by a shift in absolute tape coordinates. This relation preserves halting status and successor structure.

**Configuration embedding.** The embedding $\mathsf{Embed} : (Q \times \Gamma^{\mathbb{Z}} \times \mathbb{Z}) \to \Sigma_{\mathrm{upd}}^*$ maps a configuration $c = (q, \tau, p)$ to the canonical token sequence that "walks through" the non-blank tape region. Formally, for a tape $\tau$, define $\ell(\tau) := \min(\{0\} \cup \{i : \tau(i) \neq b\})$ and $r(\tau) := \max(\{0\} \cup \{i : \tau(i) \neq b\})$ as the left and right boundaries of the non-blank region. Then $\mathsf{Embed}(c)$ is a sequence of tokens $(q, a_i, d_i)$ where each $a_i$ is the tape symbol at position $\ell(\tau) + \sum_{j<i} d_j$ and the moves $d_i \in \{-1, 0, +1\}$ are chosen so that the sequence "walks through" the interval $[\ell(\tau), r(\tau)]$ and ends with the head aligned to $p$. By construction, $\mathsf{Update}(c_{\mathsf{blank}}, \mathsf{Embed}(c)) \sim c$. (Note: while many token sequences can produce the same configuration, $\mathsf{Embed}$ is a *deterministic* function that outputs a canonical representation.)

Since the ATM uses $O(S(n))$ space, $|\mathsf{Embed}(c)| = O(S(n))$ for all reachable configurations. Each transition tuple $\delta_i(c) = (q_i', w_i, d_i) \in \Sigma_{\mathrm{upd}}$ is a single update token. Appending $\delta_i(c)$ to $\mathsf{Embed}(c)$ yields an update sequence that represents the successor up to translation: $\mathsf{Update}(c_{\mathsf{blank}}, \mathsf{Embed}(c) \parallel \delta_i(c)) \sim \mathrm{Succ}_i(c)$. Define the canonicalization operator $\mathsf{Canon} : \Sigma_{\mathrm{upd}}^* \to \Sigma_{\mathrm{upd}}^*$ by $\mathsf{Canon}(z) := \mathsf{Embed}(\mathsf{Update}(c_{\mathsf{blank}}, z))$. Then $\mathsf{Canon}(\mathsf{Embed}(c)) = \mathsf{Embed}(c)$ and

$$\mathsf{Embed}(\mathrm{Succ}_i(c)) = \mathsf{Canon}(\mathsf{Embed}(c) \parallel \delta_i(c)). \tag{35}$$

## F.4. Transformer Construction

We now describe how the Transformer autoregressively generates tokens to implement the recursive evaluation.

**Call/return mechanism.** As in the primary proof, we use control tokens $\langle\texttt{call}\rangle, \langle\texttt{/call}\rangle, \langle\texttt{return}\rangle, \langle\texttt{/return}\rangle$ to implement recursion:

$$\mathsf{CALL}(c) := \langle\texttt{call}\rangle\mathsf{Embed}(c)\langle\texttt{/call}\rangle, \qquad \mathsf{RET}(b) := \langle\texttt{return}\rangle b\langle\texttt{/return}\rangle. \tag{36}$$

Completing $\mathsf{CALL}(c)$ pushes $\mathsf{Embed}(c)$ as a child context and removes the call block from the parent; completing $\mathsf{RET}(b)$ pops and appends $b$ to the parent.

**Evaluation transcript.** For a configuration $c = (q, \tau, p)$, we describe the step-by-step token generation. Recall that for non-halting $c$, there are exactly two applicable transitions yielding successors $c_i := \mathsf{STEP}(c, i)$ with $b_i := \mathsf{Win}(c_i)$.

For non-halting $c$, the active context cycles through three phases:

$$\mathsf{Embed}(c) \xrightarrow{\text{step 1}} \mathsf{Embed}(c) \parallel b_0 \xrightarrow{\text{step 2}} \mathsf{Embed}(c) \parallel b_0 \parallel b_1 \xrightarrow{\text{step 3}} \text{return}. \tag{37}$$

In step 1, the generator emits $\mathsf{CALL}(c_0)$, where the call payload is the canonical embedding $\mathsf{Embed}(c_0) = \mathsf{Canon}(\mathsf{Embed}(c) \parallel \delta_0(c))$; the child returns $b_0$. In step 2, similarly for $c_1$. In step 3, it emits $\mathsf{RET}(\mathsf{COMB}(c, b_0, b_1))$. We now describe each step in detail.

*Halting case:* If $c$ is halting, the context is $\mathsf{Embed}(c)$ and the generator emits $\mathsf{RET}(\mathsf{Win}(c))$.

*Non-halting case:* If $c$ is non-halting, let $c_i := \mathsf{Update}(c, \delta_i(c)) = \mathsf{Succ}_i(c)$ for $i \in \{0, 1\}$:

1. **Context:** $\mathsf{Embed}(c)$   →   **Generate:** $\mathsf{CALL}(c_0)$; child recurses and returns $b_0$.

   The generator first computes the update token $\delta_0(c) \in \Sigma_{\mathrm{upd}}$ and then emits $\mathsf{CALL}(c_0)$ whose payload is the canonical embedding $\mathsf{Embed}(c_0) = \mathsf{Canon}(\mathsf{Embed}(c) \parallel \delta_0(c))$. The child recursively evaluates $c_0$ and returns $b_0 = \mathsf{Win}(c_0)$. After return, the parent context becomes $\mathsf{Embed}(c) \parallel b_0$.

2. **Context:** $\mathsf{Embed}(c) \parallel b_0$   →   **Generate:** $\mathsf{CALL}(c_1)$; child recurses and returns $b_1$.

   Similarly, the generator emits $\mathsf{CALL}(c_1)$ with payload $\mathsf{Embed}(c_1) = \mathsf{Canon}(\mathsf{Embed}(c) \parallel \delta_1(c))$. The child returns $b_1 = \mathsf{Win}(c_1)$. After return, the parent context becomes $\mathsf{Embed}(c) \parallel b_0 \parallel b_1$.

3. **Context:** $\mathsf{Embed}(c) \parallel b_0 \parallel b_1$   →   **Generate:** $\mathsf{RET}(\mathsf{COMB}(c, b_0, b_1))$.

   With both results $b_0, b_1$ available, the generator computes $\mathsf{COMB}(c, b_0, b_1)$ (AND if $c \in Q_\forall$, OR if $c \in Q_\exists$) and emits the return block, completing the evaluation of $c$.

**Transformer construction.** It remains to verify that the next-token policy is implementable by a fixed constant-depth, constant-size Transformer with $\mathcal{O}(\log S(n))$ precision. The recursive evaluation of $\mathsf{EVAL}(c)$ reduces to the following primitive operations:

(a) Parsing the configuration embedding prefix $\mathsf{Embed}(c)$ to extract the current state $q$ (reading the state component of any update token in $\mathsf{Embed}(c)$);

(b) Halting and alternation-type detection: checking $q \in Q_{\mathrm{acc}} \cup Q_{\mathrm{rej}}$ and $q \in Q_\exists$ vs. $q \in Q_\forall$ (constant-size set membership);

(c) Computing the head position $p$ as a prefix sum of moves in $\mathsf{Embed}(c)$ (via `seq_sum`);

(d) Retrieving the scanned symbol $a = \tau(p)$ via a "rightmost match" query: find the most recent update token in $\mathsf{Embed}(c)$ that wrote to position $p$ (via `rightmost_exact_match`);

(e) Computing the successor transition $\delta_i(c) = (q_i', w_i, d_i)$ via finite lookup on $(q, a)$ (hard-coded into parameters);

(f) Computing $\mathsf{COMB}(c, b_0, b_1)$: AND/OR of returned bits based on alternation type (local gates);

(g) Canonicalization: generating the call payload $\mathsf{Embed}(c_i) = \mathsf{Canon}(\mathsf{Embed}(c) \parallel \delta_i(c))$ for $i \in \{0, 1\}$. This re-embeds the successor configuration by walking through the updated tape using (c) and (d) with the new state $q_i'$, new head position $p + d_i$, and tape symbol $w_i$ at position $p$. The output length is $|\mathsf{Embed}(c_i)| = O(S(n))$.

All primitive operations (a)–(g) above are already established in Appendix G of Yang et al. (2025b); our construction differs only in the choice of special tokens and parsing format. We refer readers to that paper for the detailed Transformer implementation.

**Conclusion.** By FASP-to-Transformer compilation (Yang et al., 2025b), the next-token rule can be implemented by a fixed constant-depth Transformer $f_\theta$ with $O(\log S(n))$ precision. The recursive model with generator $f_\theta$ decides $L$ with local space $O(S(n))$ and recursion depth $2^{O(S(n))}$. Therefore $L \in \mathsf{RM}(O(S(n)), 2^{O(S(n))})$, completing the alternative proof of Theorem 1.

## G. Proof of Theorem 2

*Proof.* Both inclusions are a direct corollary of the chain-of-thought characterization in Merrill & Sabharwal (2024). When $D = 1$ (no recursive calls), the recursive model reduces to standard autoregressive generation: the sequence grows monotonically until the model emits a return token, so the local space bound $\mathcal{O}(S(n))$ directly limits the total number of generated tokens to $\mathcal{O}(S(n))$. Setting $t(n) = \Theta(S(n))$ in their Eq. (1) and using $S(n) \geq n$ yields $\mathsf{TIME}(\mathcal{O}(S(n))) \subseteq \mathsf{RM}(\mathcal{O}(S(n)), 1)$ and $\mathsf{RM}(\mathcal{O}(S(n)), 1) \subseteq \mathsf{TIME}(\widetilde{\mathcal{O}}(S^2(n)))$, where the $\widetilde{\mathcal{O}}(\cdot)$ absorbs the polylogarithmic overhead from simulating $\mathcal{O}(\log S(n))$-precision arithmetic on a Turing machine. □

## H. Proof of Theorem 3

**Theorem 9** (Constant-Depth Recursive Models, Formal)*. For any $S(n) \geq n$, recursive models with constant recursion depth $D = O(1)$ and local space $\mathcal{O}(S(n))$ can solve any problem in* $\mathsf{SPACE}(S(n))$:

$$\mathsf{SPACE}(S(n)) \subseteq \mathsf{RM}(\mathcal{O}(S(n)), \mathcal{O}(1)). \tag{38}$$

*Moreover, for any $T : \mathbb{N} \to \mathbb{N}$,*

$$\mathsf{TM}(S(n), T(n)) \subseteq \mathsf{RM}(\mathcal{O}(S(n)), \mathcal{O}(1), \mathcal{O}(T(n))). \tag{39}$$

*Proof.* Fix any language $L \in \mathsf{SPACE}(\mathcal{O}(S(n)))$ and let $\mathsf{TM} = (\Gamma, b, Q, q_0, \delta, Q_{\mathrm{acc}}, Q_{\mathrm{rej}})$ be a deterministic single-tape Turing machine deciding $L$ using at most $c \cdot S(n)$ tape cells on inputs of length $n$. We construct a constant-size Transformer $f_{\theta(L)}$ such that the recursive model with $f_{\theta(L)}$ simulates $\mathsf{TM}$ with recursion depth $D = 2$ and local space $\mathcal{O}(S(n))$ in a time- and space-efficient manner. If this same machine also halts within $T(n)$ steps, the token-efficiency analysis below gives the strengthened membership in $\mathsf{RM}(\mathcal{O}(S(n)), \mathcal{O}(1), \mathcal{O}(T(n)))$.

**Configuration.** A *configuration* of $\mathsf{TM}$ is a triple $c = (q, \tau, p)$ where:

- $q \in Q$ is the current control state;

- $\tau : \mathbb{Z} \to \Gamma$ is the *tape contents*, a function mapping each cell index to a symbol, with $\tau(i) = b$ (the blank symbol) for all but finitely many $i$;

- $p \in \mathbb{Z}$ is the head position.

For a tape $\tau$ and position $p$, we write $\tau[p \mapsto w]$ for the tape that agrees with $\tau$ everywhere except at position $p$, where it holds symbol $w$.

**Update tokens and the update operator.** Let $\Sigma_{\mathrm{upd}} := Q \times \Gamma \times \{-1, 0, +1\}$. We interpret a token $x = (q', w, d) \in \Sigma_{\mathrm{upd}}$ as an *update*: "write $w$ at the current head cell, move by $d$, and set the control state to $q'$". For a configuration $c = (q, \tau, p)$, define

$$\mathsf{Update}(c, (q', w, d)) := (q', \tau[p \mapsto w], p + d), \tag{40}$$

and extend $\mathsf{Update}$ to sequences $x_{1:k} \in \Sigma_{\mathrm{upd}}^*$ by $\mathsf{Update}(c, x_{1:k}) := \mathsf{Update}(\mathsf{Update}(c, x_{1:k-1}), x_k)$. We also extend $\delta$ to configurations by $\delta(q, \tau, p) := \delta(q, \tau(p))$.

**Translational equivalence.** Two configurations $c_1 = (q, \tau_1, p_1)$ and $c_2 = (q, \tau_2, p_2)$ are *translationally equivalent*, written $c_1 \sim c_2$, if there exists $k \in \mathbb{Z}$ such that $\tau_1(i) = \tau_2(i - k)$ for all $i \in \mathbb{Z}$ and $p_1 = p_2 + k$. Intuitively, two configurations are translationally equivalent if they differ only by a shift in absolute tape coordinates, while their control state, tape contents, and the head's relative position within the tape are identical. This relation preserves the next update and halting status: $c_1 \sim c_2 \Rightarrow \delta(c_1) = \delta(c_2)$.

**Configuration embedding.** For a tape $\tau$, define $\ell(\tau) := \min(\{0\} \cup \{i : \tau(i) \neq b\})$ and $r(\tau) := \max(\{0\} \cup \{i : \tau(i) \neq b\})$. The embedding $\mathsf{Embed} : (Q \times \Gamma^{\mathbb{Z}} \times \mathbb{Z}) \to \Sigma_{\mathrm{upd}}^*$ maps a configuration $c = (q, \tau, p)$ to a sequence $(x_1, \ldots, x_m)$ where each $x_i = (q, a_i, d_i)$, with $a_i$ being the tape symbol at position $\ell(\tau) + \sum_{j<i} d_j$ and $d_i \in \{-1, 0, +1\}$ chosen so that the sequence "walks through" the non-blank interval $[\ell(\tau), r(\tau)]$ and ends with the head aligned to $p$. Let $c_{\mathrm{blank}} := (q_0, b^{\mathbb{Z}}, 0)$ be the blank configuration. Then $\mathsf{Update}(c_{\mathrm{blank}}, \mathsf{Embed}(c)) \sim c$. (Note: while many token sequences can produce the same configuration, $\mathsf{Embed}$ is a *deterministic* function that outputs a canonical representation; the proof only requires $\mathsf{Update}(c_{\mathrm{blank}}, \mathsf{Embed}(c)) \sim c$.)

Since $\mathsf{TM}$ is space-bounded, $|\mathsf{Embed}(c)| = \mathcal{O}(S(n))$ for all reachable configurations. Let $N := C \cdot S(n)$ for a sufficiently large constant $C$ such that $|\mathsf{Embed}(c)| \leq N$ for every reachable configuration.

**Depth-1 frame.** The depth-1 frame is the outermost frame and serves as a "dispatcher". Its role is simple: whenever its suffix matches $\langle \mathtt{call} \rangle w$ for some string $w$, it emits the closing token $\langle /\mathtt{call} \rangle$, which triggers a push of a new depth-2 frame with content $w$. This mechanism enables tail-call elimination: when the depth-2 frame returns an open call-prefix, the depth-1 frame completes the call and activates a fresh depth-2 frame.

**Depth-2 frame.** The depth-2 frame is the active simulation frame. It stores

$$\mathsf{Frame}(z, u) := z \, \| \, \langle\texttt{sep}\rangle \, \| \, u, \tag{41}$$

where $z \in \Sigma_{\mathrm{upd}}^*$ is the *summarized history* (the embedding of all past computation) and $u \in \Sigma_{\mathrm{upd}}^*$ is the *new trace* (updates generated since the last summarization). The current simulated configuration is recovered by

$$c^* := \mathsf{Conf}(z, u) := \mathsf{Update}(c_{\mathsf{blank}}, z \, \| \, u), \tag{42}$$

where the delimiter $\langle\texttt{sep}\rangle$ is ignored by $\mathsf{Update}$.

**Next-token policy (simulation vs. summarization).** Given $\mathsf{Frame}(z, u)$, let $c^* = (q^*, \tau^*, p^*) = \mathsf{Conf}(z, u)$. The next-token policy operates as follows:

(i) **Halting:** If $q^* \in Q_{\mathrm{acc}}$ (resp. $Q_{\mathrm{rej}}$), emit $\langle\texttt{return}\rangle 1 \langle/\texttt{return}\rangle$ (resp. $\langle\texttt{return}\rangle 0 \langle/\texttt{return}\rangle$) and halt.

(ii) **Simulation mode:** If $q^* \notin Q_{\mathrm{acc}} \cup Q_{\mathrm{rej}}$ and $|u| < 2N$, emit the single update token $\delta(c^*) \in \Sigma_{\mathrm{upd}}$. This appends exactly one TM step to the new trace $u$.

(iii) **Summarization mode:** If $|u| = 2N$, compute the summarized state $z' := \mathsf{Embed}(c^*)$ and emit $\langle\texttt{return}\rangle\langle\texttt{call}\rangle \mathsf{Frame}(z', \epsilon) \langle/\texttt{return}\rangle$. Under the recursive-model stack semantics, this returns an open call-prefix to the depth-1 frame, which then emits $\langle/\texttt{call}\rangle$ and activates a new depth-2 frame $\mathsf{Frame}(z', \epsilon)$.

**Correctness and resource analysis.** We now verify that the construction is correct and analyze its resource consumption: local space $\mathcal{O}(S(n))$, recursion depth 2, and token efficiency $\mathcal{O}(T)$ where $T$ is the number of TM steps.

**Correctness.** We maintain the invariant that at all times the depth-2 frame represents the current TM configuration (up to translation): after $t$ simulated steps since the last summarization, $\mathsf{Conf}(z, u) \sim c_t$, where $c_t$ is the true TM configuration after $t$ steps. The base case follows from $\mathsf{Update}(c_{\mathsf{blank}}, \mathsf{Embed}(c_0)) \sim c_0$. In simulation mode, emitting $\delta(\mathsf{Conf}(z, u))$ advances the configuration by one $\mathsf{Update}$, matching one TM transition since $\delta$ is invariant under $\sim$. In summarization mode, replacing $(z, u)$ by $(\mathsf{Embed}(\mathsf{Conf}(z, u)), \epsilon)$ preserves the represented configuration since $\mathsf{Update}(c_{\mathsf{blank}}, \mathsf{Embed}(\mathsf{Conf}(z, u))) \sim \mathsf{Conf}(z, u)$. Thus the model returns the correct accept/reject decision.

**Local space.** During simulation, the depth-2 frame has length $|z| + 1 + |u| \le N + 1 + 2N = 3N + 1 = \mathcal{O}(S(n))$. During summarization, the return payload contributes at most $|z'| + \mathcal{O}(1) = \mathcal{O}(S(n))$ additional tokens, so local space remains $\mathcal{O}(S(n))$. Thus the bound applies both to stored frames and to transient rollout outputs. The stack height is always at most 2.

**Token efficiency.** Each TM step produces exactly one emitted update token in simulation mode. A summarization happens once every $2N$ simulated steps and emits at most $|z'| + \mathcal{O}(1) \le N + \mathcal{O}(1)$ tokens. If TM halts after $T$ steps, the total number of emitted tokens is

$$T + \mathcal{O}\left(\frac{T}{2N}\right) \cdot (N + \mathcal{O}(1)) = \mathcal{O}(T), \tag{43}$$

which is linear in $T$.

**Transformer construction.** It remains to verify that the next-token policy is implementable by a fixed constant-depth, constant-size Transformer with $\mathcal{O}(\log n)$ precision. Both $\delta(\cdot)$ and $\mathsf{Embed}(\cdot)$ reduce to the following primitive operations:

(a) Parsing the summarized history $z$ and new trace $u$ (fixed-format tokenized strings);

(b) Computing the head position as a prefix sum of moves in $z \, \| \, u$ (arithmetic on $\mathcal{O}(\log S(n))$-bit integers);

(c) Retrieving the current tape symbol via a "rightmost match" query: for a given head position, find the most recent update token in $z \, \| \, u$ that wrote to that cell;

(d) A finite lookup of $\delta$ (hard-coded into parameters).

All primitive operations (a)–(d) above are already established in Appendix G of Yang et al. (2025b); our construction differs only in the choice of special tokens and parsing format. We refer readers to that paper for the detailed Transformer implementation.

If $L \in \mathsf{TM}(S(n), T(n))$, choose the witnessing Turing machine that simultaneously uses $\mathcal{O}(S(n))$ space and $\mathcal{O}(T(n))$ time. The construction above then gives recursion depth 2, local space $\mathcal{O}(S(n))$, and $\mathcal{O}(T(n))$ generated tokens, so $L \in \mathsf{RM}(\mathcal{O}(S(n)), \mathcal{O}(1), \mathcal{O}(T(n)))$. Dropping the time bound yields, for arbitrary $L \in \mathsf{SPACE}(\mathcal{O}(S(n)))$, the inclusion $L \in \mathsf{RM}(\mathcal{O}(S(n)), 2)$. $\qquad\square$

# I. Preliminaries for Section 4

## I.1. Strings

Fix a finite token alphabet $\Sigma$. We write $\Sigma^*$ for the set of all finite token strings and $|x|$ for the length of $x \in \Sigma^*$. For a length bound $L$, define $\Sigma_{\leq L} := \{z \in \Sigma^* : |z| \leq L\}$. Note that $|\Sigma_{\leq L}| \leq \sum_{i=0}^{L} |\Sigma|^i = 2^{O(L)}$.

## I.2. Polynomial-Time Generators

A generator $f$ is *polynomial-time* if there exists a deterministic Turing machine that computes $f(x)$ from $x$ in time $\mathrm{poly}(|x| + |f(x)|)$. A generator family $\mathcal{F} = (f_1, \ldots, f_k)$ is polynomial-time if each $f_\ell$ is polynomial-time.

## I.3. Oracle Turing Machine

A *deterministic oracle Turing machine* (OTM) is a deterministic multi-tape Turing machine equipped with, for each oracle name $o$ in a finite index set $\mathcal{N}$, an *oracle query tape* and an *oracle answer tape*. Each oracle is a total function $\mathcal{O}_o : \Sigma^* \to \Sigma^*$.

**Query/Answer Mechanism.** When the machine enters a distinguished *query state* $q_{\mathrm{ask},o}$, the string currently written on the oracle-$o$ query tape (from cell 0 to the first blank) is taken as the query $u$. In one transition, the oracle answer tape is overwritten with $\mathcal{O}_o(u)$ (starting at cell 0), and the machine enters a distinguished *return state* $q_{\mathrm{ret},o}$.

**Resource Measures.** Time counts ordinary TM transitions (including transitions into and out of query/return states). Space counts the number of distinct tape cells visited on *work tapes* (excluding the read-only input tape and oracle tapes).

**Relativized Complexity Classes.** For a fixed oracle family $\Omega = (\mathcal{O}_1, \ldots, \mathcal{O}_k)$:

$$\mathsf{DTIME}^{\Omega}(f(n)) = \{L : \exists\, \mathrm{OTM} \text{ with access to } \Omega \text{ deciding } L \text{ in } O(f(n)) \text{ time}\}, \tag{44}$$

$$\mathsf{DSPACE}^{\Omega}(f(n)) = \{L : \exists\, \mathrm{OTM} \text{ with access to } \Omega \text{ deciding } L \text{ in } O(f(n)) \text{ work space}\}. \tag{45}$$

**Recursive-Call Variant.** In the main text, a scaffold is formalized as an OTM-style procedure whose query names are

$$\mathsf{GEN}_1, \ldots, \mathsf{GEN}_k \qquad \text{and} \qquad \mathsf{SELF}_1, \ldots, \mathsf{SELF}_m.$$

The names $\mathsf{GEN}_\ell$ are interpreted by the given generator functions $f_\ell$. The names $\mathsf{SELF}_j$ are recursive-call interfaces; their meanings are not fixed in advance, but are solved by the least-fixpoint construction in § I.4.

**Alternating Turing Machines.** An *alternating Turing machine* (ATM) extends a nondeterministic TM by labeling each state as either *existential* ($\exists$) or *universal* ($\forall$). At an $\exists$-state, the machine accepts if *some* successor configuration accepts; at a $\forall$-state, it accepts if *all* successor configurations accept. We write $\mathsf{ASPACE}(S(n))$ for the class of languages decidable by an ATM using space $O(S(n))$.

**Space-Constructibility.** A function $S : \mathbb{N} \to \mathbb{N}$ is *space-constructible* if there exists a TM that, on input $1^n$, computes $S(n)$ in binary using $O(S(n))$ space. Common functions like $n, n^2, 2^n$ are space-constructible.

## I.4. Least-Fixpoint Semantics for Recursive Agentic Systems

We now give the formal construction of the semantics for a recursive agentic system $(\mathcal{S}, \mathcal{F})$ where $\mathcal{S} = (S_1, \ldots, S_m)$ are scaffolds and $\mathcal{F} = (f_1, \ldots, f_k)$ are generators. Each scaffold $S_i$ may issue $\mathsf{GEN}_\ell$ queries for $\ell \in \{1, \ldots, k\}$ and $\mathsf{SELF}_j$ queries for $j \in \{1, \ldots, m\}$. The former are interpreted by the known functions $f_\ell$; the latter are interpreted by the unknown partial functions being defined.

**Partial Functions and Order.** Let $\Sigma_{\perp}^* = \Sigma^* \cup \{\perp\}$, where $\perp$ denotes "undefined." This is a meta-level marker, not a string returned to or observed by a scaffold. Let $\mathcal{P} = \{F : \Sigma^* \to \Sigma_{\perp}^*\}$ be the set of partial functions. We order $\mathcal{P}$ by

*extension*: $F \sqsubseteq G$ iff for all $x \in \Sigma^*$, either $F(x) = \bot$ or $F(x) = G(x)$. The pair $(\mathcal{P}, \sqsubseteq)$ forms a complete partial order with least element $\bot_{\mathcal{P}}$ (the everywhere-undefined function). For tuples, define $\mathcal{P}^{(m)} = (\mathcal{P})^m$ with componentwise order; the least element is $\bot^{(m)} = (\bot_{\mathcal{P}}, \dots, \bot_{\mathcal{P}})$.

**One-Step Operator.** Define $\boldsymbol{\Phi}_{\mathcal{S}, \mathcal{F}} : \mathcal{P}^{(m)} \to \mathcal{P}^{(m)}$ as follows. Given $\mathbf{F} = (F_1, \dots, F_m) \in \mathcal{P}^{(m)}$, the $i$-th component $\boldsymbol{\Phi}_{\mathcal{S}, \mathcal{F}}(\mathbf{F})_i$ is defined by simulating $S_i$ on input $x \in \Sigma^*$. A query $\mathsf{GEN}_\ell(u)$ is answered by the fixed generator value $f_\ell(u)$. A query $\mathsf{SELF}_j(u)$ is answered by the current approximation $F_j(u)$ if this value is defined. If the simulation does not halt, or if it queries some $F_j(u) = \bot$, the operator value is $\bot$; in the latter case the scaffold is not given $\bot$ as an oracle answer. If the simulation halts with output $y$ without making an undefined recursive query, then $\boldsymbol{\Phi}_{\mathcal{S}, \mathcal{F}}(\mathbf{F})_i(x) = y$.

**$\omega$-Continuity and Existence.** The operator $\boldsymbol{\Phi}_{\mathcal{S}, \mathcal{F}}$ is $\omega$-continuous (Scott-continuous) on the pointed CPO $(\mathcal{P}^{(m)}, \sqsubseteq)$: each scaffold execution, if it terminates, makes only finitely many recursion queries, hence depends only on a finite stage of any increasing chain of approximants. By Kleene's fixed-point theorem, the least fixpoint $\mathbf{F}^* = \mathrm{lfp}(\boldsymbol{\Phi}_{\mathcal{S}, \mathcal{F}}) = \bigsqcup_{n < \omega} \boldsymbol{\Phi}_{\mathcal{S}, \mathcal{F}}^n(\bot^{(m)}) \in \mathcal{P}^{(m)}$ exists.

**Semantics.** The semantics of the system $(\mathcal{S}, \mathcal{F})$ is the least fixpoint $\mathbf{F}^* = (F_1^*, \dots, F_m^*)$. We identify the induced functions in the main text with these components, i.e., $\phi_i^{\mathcal{S}, \mathcal{F}} := F_i^*$. Thus $\phi_i^{\mathcal{S}, \mathcal{F}}(x)$ is the output of running scaffold $S_i$ on input $x$, where $\mathsf{GEN}_\ell$ is interpreted by $f_\ell$ and all $\mathsf{SELF}_j$ calls are resolved by the least fixpoint. If the generators in $\mathcal{F}$ are computable, then the induced functions $\phi_i^{\mathcal{S}, \mathcal{F}}$ are ordinary partial computable functions; for arbitrary generators, they are partial computable relative to the generator oracles.

### I.5. Chandra–Kozen–Stockmeyer Characterization

**Lemma 10** (Alternating-space characterization of exponential time). *For any space-constructible $S(n) \geq n$,* $\mathsf{ASPACE}(O(S(n))) = \mathsf{TIME}(2^{O(S(n))})$.

*Proof.* This is a classical result. The key insight is that an alternating TM using space $S$ has at most $2^{O(S)}$ configurations, and a deterministic simulation can explore the entire game tree in time $2^{O(S)}$ via dynamic programming. $\square$

## J. Proof of Theorem 4

*Proof of Theorem 4.* Fix $n$, an index $r \in \{1, \dots, m\}$, and input $x \in \Sigma^n$, and write $L := L(n)$ and $D := \Sigma_{\leq L}$. Let $\boldsymbol{\Phi}_{\mathcal{S}, \mathcal{F}}$ be the one-step operator from Appendix I.4. Under $L$-boundedness (Definition 4) of the evaluation of $\phi_r^{\mathcal{S}, \mathcal{F}}(x)$, every generator argument/return and every recursion argument/return that appears during evaluation lies in $D$. This $L$-boundedness assumption counts query and answer strings held by a scaffold, whereas the standard relativized space measure for oracle Turing machines in § I.3 counts only work tapes.

**Time upper bound (oracle form).** Let $\boldsymbol{\Phi}_{\mathcal{S}, \mathcal{F}}^{(\leq L)}$ be the same one-step operator as $\boldsymbol{\Phi}_{\mathcal{S}, \mathcal{F}}$, except that each scaffold simulation is run with an explicit $L$-space cutoff (counting work plus oracle tapes): if the simulation exceeds $L$ total tape cells, the corresponding approximant value is declared undefined, i.e., set to $\bot$. Since the recursive call tree of the evaluation of $\phi_r^{\mathcal{S}, \mathcal{F}}(x)$ is $L$-bounded, this cutoff never triggers on any scaffold invocation that influences $\phi_r^{\mathcal{S}, \mathcal{F}}(x)$, so $\phi_r^{\mathcal{S}, \mathcal{F}}(x)$ equals the stabilized $(r, x)$ entry of the least fixpoint of $\boldsymbol{\Phi}_{\mathcal{S}, \mathcal{F}}^{(\leq L)}$.

We compute this least fixpoint by performing Kleene iteration on the finite restriction to $D$. Define a table-valued sequence $(\phi_t)_{t \geq 0}$ where each $\phi_t$ is a tuple of partial maps $D \to D \cup \{\bot\}$ (one component per scaffold), with $\phi_0 = \bot^{(m)}$ and $\phi_{t+1} = \boldsymbol{\Phi}_{\mathcal{S}, \mathcal{F}}^{(\leq L)}(\phi_t)$ restricted to inputs in $D$. Because the restriction domain is finite and the order is by extension, each table entry can change at most once (from $\bot$ to a defined value), so the sequence stabilizes after at most $m \cdot |D|$ iterations.

To update one table entry, we simulate one one-step scaffold run on an input $q \in D$, answering generator/tool queries by oracle access to $\mathcal{F}$ and answering recursion queries $\mathsf{SELF}_j(u)$ by table lookup of $\phi_t$. By construction of $\boldsymbol{\Phi}_{\mathcal{S}, \mathcal{F}}^{(\leq L)}$, each such update halts within $\exp(O(L))$ transitions and costs $\exp(O(L))$ time. There are $m \cdot |D| = 2^{O(L)}$ entries and at most $m \cdot |D| = 2^{O(L)}$ iterations, so the total oracle-machine running time is $2^{O(L)}$. The stabilized table entry corresponding to $(r, x)$ equals $\phi_r^{\mathcal{S}, \mathcal{F}}(x)$, proving $\mathsf{DTIME}^{\mathcal{F}}(2^{O(L(n))})$.

**Eliminating the oracle.** Now assume additionally that each generator/tool in $\mathcal{F}$ is computable by a deterministic (non-oracle) TM in time $2^{O(L(n))}$ and work space $O(L(n))$ on all queries of length at most $L(n)$. We simulate the above oracle

TM by a plain TM, replacing each oracle query by running the corresponding oracle-computing TM on the query string and writing its output back before resuming the simulation. Since the oracle TM runs for at most $2^{O(L(n))}$ steps, it makes at most $2^{O(L(n))}$ oracle queries. Thus the total time is $2^{O(L(n))} \cdot 2^{O(L(n))} = 2^{O(L(n))}$, proving $\mathsf{TIME}(2^{O(L(n))})$. $\qquad\square$

## K. Proof of Theorem 5

*Proof of Theorem 5.* Fix an index $r \in \{1, \dots, m\}$. Assume the recursion stack depth is $D(n) = O(1)$ throughout evaluation of $\phi_r^{S,\mathcal{F}}(x)$ on every length-$n$ input $x$. We decide the language by directly simulating the recursive evaluation in a depth-first manner on an oracle TM with access to $\mathcal{F}$. The simulator maintains the full local configuration of the currently active scaffold simulation (including its work tapes and the bounded oracle query/answer content), and pushes/pops such configurations on a stack when encountering recursive scaffold calls and returns. By $L(n)$-boundedness, each call frame requires $O(L(n))$ space to store, and by assumption there are $O(1)$ frames simultaneously. Thus the simulation uses $O(L(n))$ work space, proving membership in $\mathsf{DSPACE}^{\mathcal{F}}(O(L(n)))$.

**Eliminating the oracle.** Now assume additionally that each generator/tool in $\mathcal{F}$ is computable by a deterministic (non-oracle) TM in time $2^{O(L(n))}$ and work space $O(L(n))$ on all queries of length at most $L(n)$. We simulate the above oracle TM by a plain TM, replacing each oracle query by running the corresponding oracle-computing TM on the query string and then resuming the simulation. Reusing $O(L(n))$ work space for each oracle computation, the overall work space remains $O(L(n))$, proving $\mathsf{DSPACE}(O(L(n)))$. $\qquad\square$

