# OpenReview forum: "Recursive Models for Long-Horizon Reasoning"
_ICML.cc/2026/Conference — ICML 2026 regular_

### Official Review · Reviewer_dV8v · 2026-02-13

**Soundness:** 1
**Presentation:** 2
**Significance:** 2
**Originality:** 3
**Overall Recommendation:** 4
**Confidence:** 4

**Summary:**

The paper focuses on theoretically analysing recursive language models. To do so, they define the class of recursive model complexity -- RMC, which is based on:
* Local space -- $S(n)$, the maximum length of a stack frame
* Recursion depth -- $D(n)$, i.e. stack depth
* Total steps -- $T(n)$, i.e. total number of generated tokens
wher $n$ is the input problem size (L197-198; right column).

A recursive lamguage model $RCM_f\in RCM$ is an extension to a base LLM $f$, such
that the LLM can call itself and return answers from such call, when generating
new tokens. When processing a recursive call **only** the current stack frame
(decided by the LLM) is retained. Once the recursive computation is complete,
(which can call itself, so that depth>1), the only the answer (and possibly
call parameters) are returned, but not the computation itself. The lowest
level of the stack $\mathbf{S}_0$ is the input sequence $x$.

For the class of RCM, the authors show:
* [MAIN RESULT] any problem that a standard Turing machine can decide in
  $\mathrm{TIME}(2^{\mathcal{O}(S(n))})$, can also be decided by an
  $RCM(\mathcal{O}(S(n)), \infty, \infty)$.
* No/Shallow recursion, even with an unlimited time (i.e. unlimited global
  context), *but limited local context* does not equal deep recursion.
  * (obvious) No recursion $\implies$ local context constraint is also a global context constraint, hence the impossibility.
  * No recursion is a subset of $\mathrm{TIME}(\mathcal{O}(S^2(n)))$
  * Constant-depth (i.e. $\mathrm{RCM}(\mathcal{O}(S(n)), \mathcal{O}(1),
    \mathcal{O}(T(n)))$) recursion subsumes $\mathrm{SPACE}(S(n))$ and
    $\mathrm{TM}(S(n), T(n))$ (turing machines working in S(n) space and T(n) time)
* RCMs are optimal. Any agentic system, even if it contains specialised LLMs,
  offers no theoretical advantage over RCMs.

The authors then proceed to validate the effectiveness of RCM. They show a
**single** experiment on a synthetic boolean satisfiability, where an LLM
trained to with recursive calls. Results show substantial outperformance of
baselines, especially in Medium-sized instances. All this is achieved whilst
keeping the local context length bounded.

**Compliance With Llm Reviewing Policy:**

Affirmed.

**Final Justification:**

weak accept

* both me and all other reviewers pointed out the limited number of experiments
* the approach only targets "workspace" (a.k.a. decomposing reasoning)

**Key Questions For Authors:**

I have left my questions/discussions in weaknesses and I gave more than 3-5 axis to improve the paper on. I will not repeat myself here. I used **boldface** for aspects I believe are worth addressing first and would lead to larger jumps in score

That being said, if a question is unclear, please reach out ASAP.

**Limitations:**

Authors have the standard statement, but no limitation is given (ctrl+F `limitation` returns nothing) in the main text.

**Strengths And Weaknesses:**

## Strengths
* The idea of having **one** LLM that can call itself, and this leading to a
  drastic improvement in capabilities has potential impact.
* The proposed RCM is simple and elegant.
* The authors have thought about non-AR models and some other scenarios, such as
  agentic workflows.
* The RCM is not less powerful than agents. Any agent system can be subsumed
  by an appropriate RCM

## Weaknesses
* **[UNSOUND ASSUMPTIONS]** (P.4, right column, L175+) "The global space
  corresponds to the total size of the current stack [...] allowing KV caches
  to be offloaded to external storage ([...]) which has **virtually unlimited
  capacity**".
  * (I will refer to CPU RAM as RAM and GPU RAM as VRAM)
  * CPU RAM is only a couple of times larger than VRAM. On a (high-end; 128GB
    RAM, 32/16 VRAM) personal device this ratio is as low as 4x-8x, on a server
    it can reach 100x. [Given that (good quality) RAM prices are on the
    rise](https://uk.pcpartpicker.com/trends/price/memory/) I would not call it
    **unlimited**. A similar argument could probably be made for disk memory.
  * **Access speeds/patterns matter.** A VRAM memory access is considerably
    faster than a RAM to VRAM transfer, [even if we focus on newer chips with a
    dedicated
    C2C](https://www.nvidia.com/en-gb/data-center/grace-hopper-superchip/).
    There the difference is ~10x (450GB/s vs 4TB/s). Also, depending on tasks
    and inputs, the amount of memory transfers (adding a stack frame, popping a
    stack frame) may be the dominating factor and outweigh the fact we will
    run the LLM on a small local context.

    Disk access, is even slower -- I am not planning to dig up a source to prove this.
* **[UNSOUND CONCLUSIONS]** (scattered around the paper). If there is anything
  that makes me choose 2 as my current score, it is this one! Take, for
  example, the statement right after Theorem 1: "This result has important
  implications as it suggests that any computable problem is modularizable,
  i.e., inherently admitting a recursive decomposition such that every subtask
  fits within a small active context". **I am not convinced this is implied
  anywhere, taking into account (6) and the paragraph after.** What you just
  proved, is that any problem a Turing machine can solve in
  $2^{\mathcal{O}(S(n))}$ **TIME** you can solve it in $\mathcal{O}(S(n))$
  local **SPACE**. Even if I assume the well known
  $SPACE(f(n))\subseteq TIME(\mathcal{O}(2^{f(n)}))$, the implications are
  that you can solve in *local* space $O(S(n))$ what can already be solved via
  TM in that amount of total space.

  Nothing in the above conclusions is related in any way to the fact whether the
  problem is modularizable or not. (A definition of what's a modularizable problem is also lacking.) Your theory is in relation to the space RCMs will
  use in their computation, i.e. **the solution** to any computable problem
  is modularizable, not the problem. Take for example needle-in-a-haystack: Given a 2M
  context, find the 10-token needle -- you don't need exponentially more than the needle additional memory, but RCMs, as currently presented, cannot avoid the 2M-length attention.

  Furthermore, In the paper $S(n)$ is not bounded from
  above by anything (please correct me if I missed it), and it might be the case
  that one can borrow the examples from the [space hierarchy theorem
  wiki](https://en.wikipedia.org/wiki/Space_hierarchy_theorem) to construct
  tasks, where the local space needed is as larger than input length as you
  want it. Every subtask/stack frame will need to
  materialise attention on sequences growing linearly with $S(n)$. There
  could exist tasks where $S(n)$ can grow very badly!

  **Please reword/improve/elaborate accordingly your conclusions/claims!**

  Similar conclusions and short refutations:
  * "any computable problem inherently admits a recursive decomposition, and
    furthermore, by doing so, the required context can be reduced
    exponentially" -- same argument as above
  * "deep recursion exponentially reduces the required attention size" -- is
    too broad of a statemtent. As a matter of fact, I don't see where the proof
    for Theorem 2 is! I presume it because that Eq. 7-8 imply that no
    recursion RCMs are equal to L, but a less theory-acquainted reader may appreciate a short note.
* *[Missing related work]* I am forgiving the authors the lack of inclusion of
  some reasonably recent preprints on recursive language models. (e.g. [this
  RLM](https://arxiv.org/pdf/2512.24601) or
  [TRMs](https://arxiv.org/pdf/2510.04871), which also have notion of
  recursion). They couldn't have captured that and I assume ~6 months grace
  period!

  However, there are works, e.g. in the field of hierarchical language models,
  ([HRM](https://arxiv.org/pdf/2506.21734) or the [Titans
  architecture](https://arxiv.org/abs/2501.00663)), which also seems to tackle
  long context scaling. Especially the latter one is ~1 year old. Will they fall in
  the same computational classes as your approach? The field includes more papers
  than that. I would appreciate a dedicated discussion on hierarchical attention/LM and long-context scaling -- currently B.3 is 6 lines in total...

* **[QUESTIONING IMPACT PRESENTAITON]** Similarly, a lot of the paper felt like
  a complexity theory one. I do not have that good knowledge of the SOTA in CS
  theory, but, to what extent are the results shown here known by complexity
  theorists? Take for example Algorithm 2, where does it utilise the fact that
  $f$ is parametrised by a neural network? Even in your proofs, you're
  constructing the transformer (Appendix F3), you're not learning it, so you
  don't have to use a transformer. This begs the question if a structure
  similar to RCM already exists. If that is *not* the case, convince us with a
  related work on complexity theory. If that is the case, I believe it is more
  scientifically correct to reword the paper, so that you propose an LLM
  inference technique that can capture the specific class.
* **[ONE SINGLE EXPERIMENT AND ONE EFFICIENCY ASPECT]** Lastly, and I hope you
  agree here, I do not think it is up to ICML standards to confirm a theory
  with a single experiment and 15 datapoints in total. Compare yourself to some
  of the more recent related works. They have multiple datasets/baselines.
  ARC-AGI can also be a good candidate dataset, because it is worth showing if
  the approach can work when exact recursion traces are available at training
  time. Similarly, Figure 3 tells only a part of the story. As a reviewer I am
  interested to see wall-clock inference time and also training times too.
* [PRESENTATION] And a last nit, I hope the presentation of 4.1 and 4.2 can be
  improved, or summarised in layman's English with hard theory left for
  appendix.

---

> ### Author Rebuttal · Authors · 2026-03-31
>
> We thank the reviewer. The core concerns are not about whether our formal results (Theorems 1–5) are correct (they are sound), but rather about (i) informal prose (e.g., "modularizable," "virtually unlimited") that could be read more broadly than the theorems state, and (ii) misunderstandings of our setting, particularly input length vs. working space and the parameterized nature of our statements. We address all below and will
> thoroughly revise according to the feedback.
>
> **UNSOUND ASSUMPTIONS**
>
> It's a practical concern, not a soundness issue. Theorem 1 is purely complexity-theoretic; none of the proofs assumes RAM-speed access or "unlimited external storage." The reviewer is objecting to implementation-level intuition for why active attention is relevant. Our results are agnostic to how suspended contexts are stored (KV cache, text or other forms). We agree "virtually unlimited" is too strong and will revise.
>
> **UNSOUND CONCLUSIONS**
>
> * **Overclaimed statement interpretation / Definition of modularity**
>
> These identify an interpretation issue, not a theorem flaw. The formal result is
> sound. To restate it precisely: for $S(n)
> \geq n$, any decision problem solvable in
> $2^{O(S(n))}$ time can be solved by a recursive
> model with a constant depth Transformer using
> $O(S(n))$ local context (Theorem 1).
>
> The reviewer correctly points out that it is not the **problem itself** that is "modularizable," but the **reasoning process** that can be organized recursively with bounded context. However, we note that nowhere in the
> paper do we define or discuss decomposing the
> input itself; our formalization is entirely about decomposing reasoning. We'll revise accordingly.
>
> * **Needle-in-a-haystack counterexample**
>
> This is not a counterexample to Theorem 1. The key distinction is between **total local context** and **workspace beyond the input**. In Theorem 1, $S(n)$ denotes the total local context available to one recursive call, including the input itself, so $S(n) \ge n$.
>
> For needle-in-a-haystack, the input length is $n$, while the additional workspace is only $O(1)$. If we denote this extra workspace by $s(n)$ (standard convention), our refined form is TIME($2^{O(s(n))}$) $\subseteq$ RCM($n + O(s(n)), \infty, \infty$), stated below Theorem 1. Here $s(n) = O(1)$, so the required context is still $n + O(1)$: the input must still be accommodated. When $s(n) \ge n$, we have $n + O(s(n)) = O(s(n))$, recovering Theorem 1.
>
> So the example is fully consistent with our
> theorem. It's a regime where input dominates and recursion yields no asymptotic benefit.
>
> * **S(n) is not bounded from above by anything**
>
> The theorem does not promise $S(n)$ is small. $S(n)$ is a parameter: for any given budget $S(n)$, deep recursion solves TIME($2^{O(S(n))}$). Tasks beyond that simply needs larger $S(n)$.
>
> * **I don't see where the proof for Theorem 2 is**
>
> Theorem 2 is from Merrill & Sabharwal (2024). We'll add an explicit pointer.
>
> **MISSING RELATED WORK**
>
> HRM/TRM are recurrent latent-state reasoners, conceptually closer to looped Transformers. Titans is a memory-augmented long-context architecture. Both are orthogonal.
>
> RLM is closer and already cited in Appendix B, but we agree it should be discussed more explicitly. Key differences: (1) RLM emphasizes input decomposition, whereas we focus on the compositionality of the reasoning process; (2) we complement theirs by providing formal analysis: RLM only considers depth $\le 2$, which we prove is no more powerful than summarization (Theorem 3) and recursion depth is critical. We'll expand related work in the revision.
>
> **QUESTIONING IMPACT PRESENTAITON**
>
> The closest classical model in expressivity is the alternating Turing machine (see Appendix G), but its design is fundamentally different. Ours is motivated by the practical constraint of context length; no such motivation previously existed.
>
> Algo 2 is intentionally architecture-agnostic. The non-trivial part is giving Algo 6-10 and showing that even a fixed constant-depth/size Transformer can simulate them (Appendices F and G). We focus on construction since this is an expressivity result; learnability is a separate question.
>
> **SINGLE EXPRIMENT**
>
> ICML encourages theory/understanding-focused submissions; comparing this paper to the experimental standards of pure empirical work would be a mispositioning.
>
> That said, we have added a Go experiment (an EXPTIME-complete problem; see our responses to 9Lja's W1 for detail). Inference time is 3.5/10.1/34.2s per instance for easy/medium/hard. Training time is in Appendix C.2 and does not depend on problem size (single model for all sizes). CoT baselines are not directly comparable as they use stronger base models or API.
>
> We evaluate on 300 held-out instances (100 per difficulty), not 15 (SATBench contains 2100 instances total, others reserved for training).
>
> **PRESENTATION**
>
> We'll improve presentation and make the limitations discussion (currently in A.3) more explicit.

---

> > ### Author Rebuttal · Reviewer_dV8v · 2026-04-01
> >
> > Thank you!
> >
> > I will adjust my score to a weak accept. Main motivations of not going further are:
> > * both me and all other reviewers pointed out the limited number of experiments, currently you only have a couple more and, unless I'm mistaken it's not a widely adopted reasoning benchmark
> > * your approach only targets "workspace" (a.k.a. decomposing reasoning). Sometimes the reasoning has to be performed on large amounts of input.

---

### Official Review · Reviewer_fBUd · 2026-02-25

**Soundness:** 4
**Presentation:** 3
**Significance:** 3
**Originality:** 2
**Overall Recommendation:** 4
**Confidence:** 3

**Summary:**

The paper extends a recent idea using function stacks to dynamically collapse the context window of LLMs, enabling longer reason chains without blasting memory limits. Specifically, the authors follow the idea of letting the LLM call itself with a particular argument string. The context is initialized to just this argument string for the recursive calls, while only the answer of the recursion is added to the context of the caller, without potential intermediate reasoning tokens. This differs from prior work in not persisting the entire context for each recursive call. The authors show that theoretically the scheme allows solving problems requiring an exponential number of steps 2^O(n) with only a polynomially sized context window O(n). Note that this significant improvement over the standard CoT approach, which naturally requires a context window that coincides with the computation steps. The authors show empirically the potential of this approach on the SAT problem, where it is expected that the algorithm requires an exponential number of steps in the size of the problem.

**Compliance With Llm Reviewing Policy:**

Affirmed.

**Final Justification:**

The rebuttal answered my questions. My overall evaluation remains positive, as before. The paper addresses the important problem of LLM-based reasoning. They show a technique to improve these capabilities at the same time as limiting the context size. Weaknesses in the empirical evaluation will be addressed in the camera ready, according to the authors' response.

**Key Questions For Authors:**

Have you compared to other approaches, summarization or particularly that by Yang et al? In the particular approach considered in the experiments, you lose the decisive benefit of decoupling recursion depth with the context window. The results are hence a bit misleading, and a comparison to other baselines (beyond CoT) is really needed to really understand the results and potential benefits.

**Limitations:**

Yes

**Strengths And Weaknesses:**

The context window size is one of the big limitations in current LLM architectures; and especially for complex reasoning tasks. At the same time, during the process of autoregressive generation, the context gets polluted with irrelevant tokens. The paper presents an interesting approach to address those deficiencies using a classic method in computer science: recursion. The paper presents a thorough theoretical analysis showing the benefits of apply recursion to LLM reasoning. The paper is overall well written. Theorems and statement seem to be sound.

I mainly have two critique points. First, the delta of the presented idea with respect to the earlier work by Yang et al 2025b is relatively small. Yang et al already considered recursion, yet as opposed to this work, they always maintained the entire context in each recursive call. The sole trick, but is an important trick, in this work is to allow independent contexts for each recursive call. Only this is what gives rise to exponential separation result, where one can solve problems with 2^O(n) steps using only an O(n) context window.

The maybe bigger limitation is the experimental evaluation. The experiments lack important baselines, in particular that of the earlier work by Yang et al 2025b. Moreover, more benchmarks should be considered to understand the benefits of the presented method in different settings.

Minor point 1: that all computable functions are recursively computable is a classic CS result and should be acknowledged as such.

Minor point 2: I found the definition discussion of the relation agentic systems not very interesting. First, it is quite trivial to see that multi "scaffolds" can trivially be represented by a single scaffold using a simple switch case. On the other hand, the considering of multiple scaffolds however significantly increases the complexity of the interpretation of the construction, e.g., necessitating arguing about fixed points. To be honest, I do not really see the point of this whole construction. In my opinion, the space would be much better dedicated to a more exhaustive empirical evaluation.

---

> ### Author Rebuttal · Authors · 2026-03-31
>
> **Weakness 1: "The delta of the presented idea with respect to the earlier work by Yang et al 2025b is relatively small. Yang et al already considered recursion, yet as opposed to this work, they always maintained the entire context in each recursive call. The sole trick, but is an important trick, in this work is to allow independent contexts for each recursive call."**
>
> We appreciate the reviewer recognizing context isolation as "an important trick." In fact, this is exactly the source of the exponential separation: we prove that Yang et al.'s mechanism is equivalent to constant-depth recursion, which with local context $S(n)$ can solve at most SPACE($S(n)$) (Theorems 3 and 5). Context isolation enables deep recursion, which reaches TIME($2^{O(S(n))}$) with the same local context (Theorem 1). With the practice of context isolation becoming increasingly prevalent in modern agentic systems, understanding what it can bring in terms of computational power is an important question, and we believe our results provide a formal answer that may also be of independent interest in classical complexity theory.
>
> **Weakness 2 & Q1: "The experiments lack important baselines, in particular that of the earlier work by Yang et al 2025b. Moreover, more benchmarks should be considered." / "Have you compared to other approaches, summarization or particularly that by Yang et al? In the particular approach considered in the experiments, you lose the decisive benefit of decoupling recursion depth with the context window. The results are hence a bit misleading, and a comparison to other baselines (beyond CoT) is really needed."**
>
> We have supplemented a new experiment on Go (a EXPTIME-complete probelm, theoretically beyond the reach of both CoT and PENCIL (Yang et al. 2025b)), where we compare our recursive model against both baselines. Our recursive model significantly outperforms both, consistent with the theoretical prediction. See our response to Reviewer 9Lja's Weakness 1 for full results.
>
>
> **Minor Point 1: "That all computable functions are recursively computable is a classic CS result and should be acknowledged as such."**
>
> We agree. We discuss the connection to classical recursion theory in Appendix A but will make this more explicit in the main text.
>
> **Minor Point 2: "I found the definition discussion of the relation agentic systems not very interesting. The considering of multiple 'scaffolds' significantly increases the complexity of the interpretation of the construction, e.g., necessitating arguing about fixed points. The space would be much better dedicated to a more exhaustive empirical evaluation."**
>
> We appreciate the feedback. One purpose of this section is to demonstrate that recursion can be realized in more flexible ways in practice (heterogeneous models, tools, mutual recursion among agents, etc.), all achieving the same computational power. That said, we agree the presentation can be improved and will do so in the revised paper.

---

> > ### Author Rebuttal · Reviewer_fBUd · 2026-04-02
> >
> > Thank you for your response. My questions have all been addressed, leaving my rating as is. The Go results look interesting and promising. Extending the experiments accordingly would definitely strengthen the paper.

---

### Official Review · Reviewer_GjhD · 2026-03-13

**Soundness:** 3
**Presentation:** 3
**Significance:** 3
**Originality:** 3
**Overall Recommendation:** 4
**Confidence:** 3

**Summary:**

Motivated by the limited attention window constraint of LLMs, this paper proposes a recursive model approach to overcome the bottleneck of long-horizon reasoning. Specifically, the model recursively invokes itself to solve subtasks within contextually isolated sequences. The authors theoretically prove that such recursive decomposition requires exponentially smaller active context than standard autoregressive models, strictly surpassing single-context management methods like summarization. Empirically, a 3B model trained with this recursive reasoning paradigm significantly outperforms frontier LLMs on complex SAT problems.

**Compliance With Llm Reviewing Policy:**

Affirmed.

**Final Justification:**

The rebuttal resolved all of my concerns, and I appreciate the authors’ careful clarifications. However, these responses do not substantially change my overall evaluation of the paper. I therefore maintain my final rating as Weak Accept.

**Key Questions For Authors:**

1.Given that the empirical validation is restricted to a single domain, how do the authors ensure the proposed method generalizes across diverse tasks? Could you provide results on additional benchmarks to robustly support the paper's claims?

2.Does the proposed method introduce an "alignment tax" or cause catastrophic forgetting? Please provide evaluation metrics on standard, unrelated benchmarks to prove that addressing the target problem does not negatively impact the LLM's general capabilities.

3.This method involves discarding intermediate reasoning steps of subtasks, which introduces a significant risk of compounding errors. That means, an incorrect output in one subtask could lead to a cascading failure of the entire workflow. How does the model mitigate error propagation? Can the authors provide further ablation studies or robustness analyses (e.g., measuring fault tolerance) to address this concern?

**Limitations:**

yes

**Strengths And Weaknesses:**

Strengths:

1.Through fine-tuning alone, the authors enable a small-parameter model to outperform much larger models.

2.This paper provides rigorous and comprehensive theoretical proofs.

3.This paper is well written and easy to follow.

Weaknesses:

1.The empirical evaluation is relatively limited in scope. Consequently, it remains unclear whether the observed performance gains can generalize to a broader range of downstream tasks.

2.The paper lacks sufficient evaluation to confirm that the proposed method does not inadvertently degrade or compromise the model's pre-existing general capabilities.

---

> ### Author Rebuttal · Authors · 2026-03-31
>
> **Weakness 1 & Q1: "The empirical evaluation is relatively limited in scope. It remains unclear whether the observed performance gains can generalize to a broader range of downstream tasks." / "How do the authors ensure the proposed method generalizes across diverse tasks? Could you provide results on additional benchmarks to robustly support the paper's claims?"**
>
> Theoretically, we have shown the advantage of recursive models is not tied to specific downstream tasks: Theorem 1 proves that any decision problem solvable in $2^{O(S(n))}$ time admits a recursive decomposition using only $O(S(n))$ local context, which is an exponential improvement over standard CoT (Theorem 2).
>
> Indeed, our empirical evaluation has not yet covered a broad range of tasks. Extending to domains without known recursive algorithms would require training models to discover recursive strategies on their own, e.g., through RL, which introduces substantial new challenges (combinatorially larger exploration space, harder credit assignment across nested calls) that no prior work has addressed. We view this as a promising direction that is beyond the scope of this paper but well motivated by our theoretical framework, and will discuss it in the revised paper.
>
> That said, we have added a new experiment on Go (EXPTIME-complete, adversarial game-tree search), which is qualitatively different from SAT (NP-complete, backtracking search). Our recursive model significantly outperforms CoT on both tasks. See our response to Reviewer 9Lja's Weakness 1 for full details.
>
> **Weakness 2 & Q2: "The paper lacks sufficient evaluation to confirm that the proposed method does not inadvertently degrade or compromise the model's pre-existing general capabilities." / "Does the proposed method introduce an 'alignment tax' or cause catastrophic forgetting? Please provide evaluation metrics on standard, unrelated benchmarks to prove that addressing the target problem does not negatively impact the LLM's general capabilities."**
>
> This is a good question. We do not claim preservation of general capabilities in our current task-specific SFT setting—this is a well-known limitation of task-specific fine-tuning and not unique to recursive models. Our experiments are proof-of-concept demonstrations that recursive reasoning can be learned effectively. A more principled approach would be to train the model to recurse across diverse domains via RL, which would better preserve general capabilities. As discussed above, this is beyond the scope of this paper and we leave it as future work.
>
> **Question 3: "This method involves discarding intermediate reasoning steps of subtasks, which introduces a significant risk of compounding errors. An incorrect output in one subtask could lead to a cascading failure of the entire workflow. How does the model mitigate error propagation? Can the authors provide further ablation studies or robustness analyses (e.g., measuring fault tolerance) to address this concern?"**
>
> We discuss error accumulation in Section A.3. Error accumulation does increase with recursion depth: our results show accuracy dropping from 98% (easy) to 64% (hard) as deeper search trees are needed. However, CoT suffers from the same issue with long trajectories, and in fact performs substantially worse across all difficulty levels. Recursive models also partially mitigate this by discarding intermediate reasoning upon return, so errors within a subtask do not pollute sibling or parent computations.

---

> > ### Author Rebuttal · Reviewer_GjhD · 2026-04-02
> >
> > My concerns have been resolved, and I maintain my rating.

---

### Official Review · Reviewer_9Lja · 2026-03-13

**Soundness:** 2
**Presentation:** 3
**Significance:** 3
**Originality:** 3
**Overall Recommendation:** 4
**Confidence:** 1

**Summary:**

This paper proposes recursive models as a minimal mechanism for long-horizon reasoning in LLMs. The core idea is to equip a base LLM with \<call\> and \<return\> tokens enabling self-invocation in isolated contexts. The paper proves that deep recursion exponentially reduces required context compared to single-context strategies, and shows that any recursive agentic workflow achieves exactly the same computational power as this minimal design. Experiments on SAT demonstrate strong empirical performance of a fine-tuned 3B model over frontier LLMs.

**Compliance With Llm Reviewing Policy:**

Affirmed.

**Key Questions For Authors:**

1. The training data is generated by running DPLL — how would you obtain recursive training traces for tasks where no efficient recursive algorithm is known? Is the approach fundamentally limited to algorithmically structured problems?
2. How does accuracy degrade as a function of recursion depth on harder instances? Is there empirical evidence of error accumulation across recursive calls?

**Limitations:**

No. Should experiment on more tasks in more domains.

**Strengths And Weaknesses:**

Strengths
1. The paper provides a clean formalization of recursive models, with well-defined complexity classes (RCM) and sharp separations between deep recursion (EXPTIME), constant-depth recursion, and summarization (PSPACE). The hierarchy is easy to follow.
2. The paper makes a convincing case that two tokens suffice for optimal recursive computation, which has direct architectural implications.
3. The paper is well-structured and well-written.


Weaknesses
1. Single-task evaluation. All experiments are on SAT. There is no evidence the approach generalizes to tasks without an obvious recursive algorithm available for trace generation.
2. Training data relies on algorithm distillation. Reasoning traces are generated by running DPLL, not by the model discovering recursion itself. It is unclear whether this extends to domains without a known efficient recursive solver.

---

> ### Author Rebuttal · Authors · 2026-03-31
>
> **Weakness 1: "Single-task evaluation. All experiments are on SAT. There is no evidence the approach generalizes to tasks without an obvious recursive algorithm available for trace generation."**
>
> We have supplemented a new experiment on the game of Go, which is EXPTIME-complete and thus requires exponentially long reasoning trajectories. We compare three approaches: CoT, PENCIL (Yang et al. 2025b), a summarization-based model whose expressiveness is theoretically bounded by PSPACE, and our recursive model. For faster evaluation, accuracy is measured as full-trajectory exactness (every token must be correct), which is a lower bound on end-to-end accuracy. Due to the limited rebuttal period, these are preliminary results on 4×4 Go (Japanese rules), trained on 50K trajectories and validated on 5K. We report teacher-forcing validation accuracy at 15K and 30K training steps:
>
> | | CoT | PENCIL | Recursive (ours) |
> |---|---|---|---|
> | 15K steps | 39.5% | 47.2% | 87.2% |
> | 30K steps | 60.1% | 63.0% | 93.7% |
>
> Our recursive model significantly outperforms both baselines, consistent with the theoretical prediction that Go (EXPTIME-complete) is beyond the reach of single-context approaches. Context length statistics (over validation set):
>
> | | Mean | P0 | P25 | P50 | P75 | P99 | Max |
> |---|---|---|---|---|---|---|---|
> | RCM (max frame) | 89 | 54 | 82 | 82 | 96 | 124 | 138 |
> | CoT (trace len) | 2,020 | 668 | 1,004 | 1,484 | 2,540 | 7,320 | 10,020 |
> | PENCIL (active ctx) | 233 | 149 | 188 | 224 | 266 | 404 | 518 |
>
> More comprehensive results (e.g. different problem sizes) and experimental details (e.g. how trajectories are generated) will be included in the revised paper.
>
> For extending to the case where no obvious recursive algorithm available, see our answer below.
>
> **Weakness 2 & Q1: "Training data relies on algorithm distillation. Reasoning traces are generated by running DPLL, not by the model discovering recursion itself. It is unclear whether this extends to domains without a known efficient recursive solver." / "How would you obtain recursive training traces for tasks where no efficient recursive algorithm is known? Is the approach fundamentally limited to algorithmically structured problems?"**
>
> We agree that a more general and interesting direction is to let the model itself discover how to decompose a given problem using recursive calls. A natural approach is reinforcement learning, for example: given problems with verifiable answers, one can instruct the model to solve them using recursive calls, sample multiple recursive reasoning trajectories, and use outcome-based rewards (e.g., correctness of the final answer) to reinforce successful decomposition strategies via policy optimization.
>
> However, RL for recursive reasoning is substantially more challenging than standard CoT RL: the model must explore a combinatorially larger space (deciding when to call, what to delegate, and when to return), and reward assignment across nested recursive calls is significantly harder than in CoT reasoning. To our knowledge, no existing work has successfully trained models for deep recursion (depth $> 2$), and we believe this is an open problem in its own right. This paper focuses on establishing the theoretical foundations and preliminary empirical investigation of recursive models; we will discuss this direction in the revised paper.
>
> **Question 2: "How does accuracy degrade as a function of recursion depth on harder instances? Is there empirical evidence of error accumulation across recursive calls?"**
>
> Yes, harder SAT instances require deeper search trees and thus deeper recursion, which makes errors more likely to accumulate. This is reflected in our main results (Table 1): accuracy drops from 98% (easy) to 95% (medium) to 64% (hard).
>
> However, this concern is not unique to recursive models: standard CoT with long trajectories also accumulates errors at intermediate steps, and in fact performs substantially worse across all difficulty levels. Moreover, as discussed in Section A.3, recursive models offer a partial mitigation that CoT lacks: upon return, intermediate reasoning within a subtask is discarded, so errors made there do not pollute sibling or parent computations.

---

> > ### Author Rebuttal · Reviewer_9Lja · 2026-04-03
> >
> > Thanks for resolving my questions. I would like to keep the current (positive) score.

---

### Decision · Program_Chairs · 2026-04-30

**Decision:**

Accept (regular)

**Comment:**

All reviewers raise concerns regarding the limited experimental evaluation. At the same time, they appreciate the clean formalization of recursive models and the theoretical results on complexity classes.

Overall, I find the current level of empirical evidence acceptable for this submission, given its strong theoretical contributions. The paper provides a clean and well-motivated framework for recursive models, which constitutes a meaningful contribution to the machine learning community.